# *Taenia solium* TAF6 and TAF9 bind to a downstream promoter element present in the *Tstbp1* gene core promoter

Oscar Rodríguez-Lima[1], Ponciano García-Gutiérrez[2]\*, Lucía Jiménez[1], Laura A. Velázquez-Villegas[3], Angel Zarain-Herzberg[4], Roberto Lazzarini[5], Karel Estrada[6], Abraham Landa[1]\*

**1** Facultad de Medicina, Departamento de Microbiología y Parasitología, Universidad Nacional Autónoma de México, Ciudad de México, México, **2** Departamento de Química, Universidad Autónoma Metropolitana-Iztapalapa, Ciudad de México, México, **3** Departamento de Fisiología de la Nutrición, Instituto Nacional de Ciencias Médicas y Nutrición Salvador Zubirán, Ciudad de México, México, **4** Facultad de Medicina, Departamento de Bioquímica, Universidad Nacional Autónoma de México, Ciudad de México, México, **5** Departamento de Biología Experimental, Universidad Autónoma Metropolitana-Iztapalapa, Ciudad de México, México, **6** Instituto de Biotecnología, Universidad Nacional Autónoma de México, Morelos, México

\* landap@unam.mx (AL); pgarcia@xanum.uam.mx (PG-G)

**Data Availability Statement:** All relevant data are within the manuscript and its Supporting Information files.

## Abstract

Transcription regulation in cestodes has been little studied. Here, we characterize the *Taenia solium* TATA-binding protein (TBP) gene. We found binding sites for transcription factors such as NF1, YY1, and AP-1 in the proximal promoter. We also identified two TATA-like elements in the promoter; however, neither could bind TBP. Additionally, we mapped the transcription start site ($A_{+1}$) within an initiator and identified a putative downstream promoter element (DPE) located at +27 bp relative to the transcription start site. These two elements are important and functional for gene expression. Moreover, we identified the genes encoding *T. solium* TBP-Associated Factor 6 (TsTAF6) and 9 (TsTAF9). A Western blot assay revealed that both factors are expressed in the parasite; electrophoretic mobility shift assays and super-shift assays revealed interactions between the DPE probe and TsTAF6-TsTAF9. Finally, we used molecular dynamics simulations to formulate an interaction model among TsTAF6, TsTAF9, and the DPE probe; we stabilized the model with interactions between the histone fold domain pair in TAFs and several pairs of nucleotides in the DPE probe. We discuss novel and interesting features of the TsTAF6-TsTAF9 complex for interaction with DPE on *T. solium* promoters.

## Introduction

The transcription of eukaryotic protein-coding genes is regulated by RNA polymerase II (Pol II) and a group of proteins known as general transcription factors (TFs), which include TFIIA, TFIIB, TFIID, TFIIE, TFIIF, and TFIIH. These factors collectively constitute the pre-initiation complex (PIC). TFIID, the first to bind the core promoter, acts as the scaffold for the assembly of the other transcription complexes and positions the RNA polymerase correctly. TFIID

**Funding:** This work was supported by Dirección General de Asuntos del Personal Académico, UNAM (DGAPAPAPIIT-IN205422 granted to AL and PAPIIT-IA207124 granted to ORL), and Dirección de Cómputo y de Tecnologías de Información y Comunicación (Miztli, LANCAD-UNAM-DGTIC-344). The funders had no role in study design, data collection and analysis, decision to publish, or preparation of the manuscript.

**Competing interests:** The authors declare that they have no conflict of interest.

comprises the TATA-binding protein (TBP) and a set of highly conserved TBP-associated factors (TAFs) across different species [1].

The core promoter is the minimum nucleotide sequence required for transcription initiation in a gene, and it contains well-defined elements in various organisms. Three of these elements—the TATA-box, the initiator element (Inr), and the downstream promoter element (DPE)—are particularly crucial for gene transcription [1]. The Inr is a distinctive element located at the core promoter that includes the transcription start site (TSS). It is characterized by the consensus sequence $YYA_{+1}NWYY$ in mammalian cells, where $A_{+1}$ denotes the TSS [2]. TAF1 and TAF2 are known to bind the Inr to form a stable complex with TBP [3,4]. The TATA-box, identified as TATAWAAR in mammals, is positioned approximately 24–31 base pairs upstream of the TSS and recognized by TBP [2]. It is present in 43% of *Drosophila* spp core promoters [5] and in 10–20% of human promoter genes [6].

The DPE, discovered in *Drosophila* TATA-box-deficient promoters, is situated 28–32 base pairs downstream of the TSS [7] and has a consensus sequence of RGWYV. TAF6 and TAF9 bind to the DPE [8–10]. When present on its own, the Inr within a core promoter exhibits a notably low transcription rate by Pol II. Nevertheless, when integrated with six specificity protein 1 (Sp1) binding sites within a synthetic core promoter it facilitates substantial transcription levels [11]. As demonstrated in a synthetic promoter that includes the TATA-box, Inr, and DPE elements (referred to as the super core promoter, SCP), there is an increase in both affinity and transcription levels [12]. In nature, the presence and variation of these three elements in core promoters influences the strength of gene transcription [5].

The functions of TAFs are myriad and encompass basal transcription [13], co-activation [14], and promoter recognition [10]. They thereby aid in the assembly and initiation of transcription. Biochemical and structural analyses have revealed that TAFs harbor histone-like motifs, such as the histone fold domain (HFD), which is pivotal for the heterodimerization of many TAFs. Nonetheless, they also possess evolutionarily conserved noncanonical features distinct from histones, the functions of which largely remain enigmatic in higher organisms and cestodes [15,16]. On the other hand, advanced structural techniques, such as X-ray crystallography, cryo-electron microscopy (cryo-EM), and biochemical studies, have provided insights into the special organization of TBP, TAFs, Pol II, and other components within the PIC. Such work has enhanced our understanding of the function and interactions of TF [17,18].

Unlike more commonly studied model organisms, cestodes are characterized by unique life cycles and genomic architectures that complicate genetic and molecular analyses. Investigating the transcriptional mechanisms in cestodes is crucial for several reasons. To begin with, such work could reveal novel biological pathways and mechanisms unique to parasitic life forms, which could in turn enhance our understanding of parasite biology and evolution. Secondly, a deeper understanding of cestode transcription could identify potential targets for therapeutic interventions, crucial for combating infections such as cysticercosis and hydatid disease, which pose significant health burdens worldwide. Lastly, studying these systems can provide insights into the adaptation of organisms to parasitism, conferring new knowledge useful for parasitology and comparative genomics.

We previously reported a cDNA encoding *Taenia solium* TBP1 (TsTBP1), which binds to the TATA-box of the core promoter of the Ts2-CysPrx (*TsPrx*) and actin 5 (*pAT*5) genes and localizes in the nucleus of larval stage (henceforth cysticerci, [19]). In this study, we cloned and characterized the TATA-binding protein 1 gene of *Taenia solium* (*Tstbp1*) to reveal cis and trans elements on its core promoter, the presence of TsTAF6 and TsTAF9 in the nucleus of cysticerci, and their interaction with the DPE of *TsTbp1*. Moreover, molecular dynamics revealed that TsTAF6 and TsTAF9 form an HFD capable of creating a TAF6-TAF9 complex that interacts with the DPE.

## Materials and methods

### Biological samples

We obtained *T. solium* cysticerci from meat of naturally infected pigs sourced from rural slaughterhouses in Mexico approved by the Ministry of Health of México. The cysticerci were washed three times with sterile phosphate-buffered saline (PBS) and stored at −70°C until they were analyzed. *T. solium* adults were obtained from the small intestines of experimentally infected hamsters, as previously described [20]. Briefly, the hamsters were fed with meat infected with cysticerci and after 8 weeks they were euthanized. The adult parasites were retrieved from the small intestine and washed as before.

### Ethics statement

Experimentally infected hamsters were maintained in the animal research facility from School of Medicine, UNAM, for 8 weeks at constant temperature of 22°C, with light/obscurity cycles of 12/12 hours, and food and water *ad libitum*. For the anesthesia and analgesia of the animals, before euthanasia, we used 100 mg/kg of ketamine plus 10 mg/kg of xylazine via IP. Euthanasia was performance by administration of sodium pentobarbital (200 mg/kg) via IV in jugular vein, in accordance with Official Mexican Norms: NOM-062-ZOO-1999 for the production, care, and use of laboratory animals. Rigor mortis was confirmed before dissection of animals. Pork with cysticerci was acquired for research purposes in rural slaughterhouses in endemic areas of swine cysticercosis in Mexico, which are approved by the Mexican Ministry of Health and use protocols based on the NOM-033-SAG/ZOO-2014 for methods for killing domestic and wild animals. The research protocol was approved by the Research and Ethics Committee of the School of Medicine at the Universidad Nacional Autónoma de México (007–2012).

### Gene isolation

We prepared the *T. solium* genomic DNA and screened 100,000 clones from a λZAPII genomic DNA library of *T. solium* cysticerci using the methodology of Campos et al. [21]. We use a full-length cDNA probe encoding TsTBP1 [19]. The clones that we obtained (~1.5 kb) were amplified using a primer from the λZAPII vector, purified, and cloned into the pCRII vector (Invitrogen by Thermo Fisher Scientific, USA), following the manufacturer's instructions. The plasmids were sequenced on an ABI Prism model 373 automated DNA sequencer (Applied Biosystems, USA). We used PCGENE (Intelligenetics, USA) and Clustal X software (http://www.clustal.org/clustal2/) for the nucleotide sequences analysis and multiple alignments; we used the Patch 1.0 BioBase server (http://www.gene-regulation.com) for the core promoter element analysis.

### Transcription start site determination

We extracted total RNA from *T. solium* cysticerci using TRIzol (Invitrogen); the RNA then served as a template for TSS determination, which was accomplished using a Smart 5′-RACE cDNA Amplification Kit (Clontech, USA). The 5′-RACE fragments were PCR- amplified using the reverse primer TBPRE (inner, 5′-TCTTATCTCAAGATTTACTGTACACAC-3′), TBPRE2 (outer, 5′-CACAATGTTTTGTAGCTGTGGCTGAGG-3′), the forward primer SMARTII (5′-AAGCAGTGGTATCAACGCAGAGTACGCGGG-3′), following the manufacturer's directions. The resulting amplified bands were cloned into pCRII (Invitrogen) and sequenced as described above.

## Southern and northern blot analysis

For Southern blot analysis, 10 μg of genomic DNA was digested separately with *Hind*III, *BamH*I, *EcoR*I, and *Bgl*II, resolved on a 1% agarose gel, and blotted on a nylon membrane (Amersham, USA). For the northern blot analysis, 10 μg of total larval RNA from *T. solium* cysticerci was resolved by electrophoresis on a 1% agarose gel with formaldehyde and blotted onto a nylon membrane.

For both blots, a probe of full cDNA for TsTBP1 labeled with Random Primer kit using α-$^{32}$P (Roche, Switzerland). Prehybridization and hybridization were performed as previously described [22]. Briefly, the samples were incubated overnight at 55˚C in Denhart's solution 5X and 0.5% SDS. We next washed the samples four times in Denhardt's solution 2X and 0.1% SDS; a final wash was done using Denhart's solution 1X plus 0.01% SDS at 58˚C. Finally, the marked bands were revealed by exposing the membrane on X-ray films (Kodak, USA) for 24 h.

## Relative expression of *TsTbp1*

For the real-time PCR, 3 μg of total RNA from cysticerci and adult *T. solium* was reverse transcribed to cDNA using SMARTScribe Reverse Transcriptase (Clontech, USA), following the manufacturer's guidelines. Next, 200 ng of cDNA was used in a 10 μl reaction volume. We used the following primers: TBP-X1 (5´- ATGCAGCCAACCCCCATCAATCAG-3´) and TBP-X2 (5'-TTAGCCAGTAAGTGCTGG-3´) for *Tstbp1* gene. *T. solium* Cu,Zn superoxide dismutase (*TsCu,Zn-SOD*) gene amplification was conducted using the primers SOD-6 (5´-AAGCACG GCTTTCACGTCC-3´) and SOD-2 (5´-ACGACCCCCAGCGTTGCC-3´). The reactions were carried out in a PowerUp SYBR Green Master Mix in the Step One Real-Time PCR System (Applied Biosystems, USA). The PCR scheme used was 95˚C for 10 min 40 cycles of the following program: 95˚C for 15 sec, 60˚C for 1 min, and 72˚C for 30 sec. The *TsTbp1* mRNA levels were normalized to the *TsCu,Zn-SOD* gene, with relative quantities calculated using the comparative CT method [23].

## Plasmid constructions and luciferease reporter assay

The core promoter region of *Tstbp1* (containing Inr and DPE) and two mutated versions of the core promoter (DPE-less and Inr-and DPE-less) were cloned into a pGL4.10 vector (Promega, USA) using the *Nhe*I and *Hind*III restriction sites. To evaluate luciferase activity, we grew HEK-293 cells in DMEM supplemented with 10% (vol/vol) FBS, penicillin (100 U/mL), and streptomycin (100 μg/mL) at 37˚C in a 5% (vol/vol) $CO_2$ incubator. HEK-293 cells (180,000 cells per well) were plated in 12-well plates in DMEM without phenol red and supplemented with 5% (vol/vol) FBS treated with charcoal and dextran 24 h prior to transfection. For the assays, 500 ng of each luciferase reporter gene construct was cotransfected with 10 ng of pRL-TK Renilla expression vector (Promega) using Lipofectamine 3000 (Invitrogen). Twenty-four hours after transfection, we lysed the cells to measure luciferase activity using the Luciferase Assay Dual System; we also obtained luminometric measurements using a Veritas microplate luminometer (Turner Biosystems, USA).

## cDNAs isolation of TsTAF6 and TsTAF9

The cDNA encoding TsTAF6 and TsTAF9 was obtained from the *T. solium* genome project (*Taenia solium*_PRJNA170813 [24]). We used PCR to amplify the sequences of these factors with *T. solium* cysticerci cDNA as a template and oligonucleotides designed from the start and end of each TAF6 and TAF9 (S1 and S2 Figs). The primer sequences included TsTAF6F (5 ´ATGTTTTCGGAAGAGCGAAAG-3'), TsTAF6R (5´-CTAGTGGGAGACATTGAGGG-3'),

TsTAF9F (5'-ATGGACGGGTTCGAGCAACG-3'), and TsTAF9R (5´CTAAAATCCTTCAATA AGGTAAGG-3'). The PCR conditions were as follows: initial denaturation for 3 min at 95˚C; 30 cycles of 1 min at 95˚C, 1 min at 56˚C and 2 min at 72˚C; and a final extension step for 5 min at 72˚C. The amplified products were cloned into the pCRII vector (Invitrogen), sequenced, and analyzed using PCGENE software (Intelligenetics).

## Localization of TsTAF6 and TsTAF9 by western blot and immunofluorescence

Nuclear extracts from *T. solium* and HEK-293 cells were prepared as described previously [19,25]. For the western blot, 5 μg of nuclear extract per mm of gel was transferred to PVDF membranes. The membranes were incubated with rabbit anti-*T. solium* TBP1 N-terminal, rabbit anti-human TBP (ab28175, Abcam, USA), rabbit anti-human TAF6 (HPA006566, Sigma-Aldrich, USA), mouse anti-human TAF9 (sc271463, Santa Cruz, USA) antibodies, and normal rabbit IgG in a 1:100 dilution and washed with PBS-0.3% Tween and incubated with peroxidase-conjugated anti-rabbit IgG. Bound antibodies were revealed using 3,3′-diaminobenzidine and 1% $H_2O_2$.

For the immunofluorescence analyses, *T. solium* cysticerci were embedded in tissue-freezing medium (Triangle Biomedical Science, USA), frozen in liquid nitrogen, and stored at -70˚C. Frozen sections 6–8 μm thick were prepared and fixed with 4% paraformaldehyde in PBS. The samples were permeabilized with 0.01% (v/v) Triton-X 100 for 30 min and blocked with 3% (w/v) BSA in PBS for 30 min. The sections were then incubated overnight with rabbit anti-human TAF6 and mouse anti-human TAF9 antibodies. After being rinsed three times with PBS, the sections were incubated for 60 min at room temperature with Alexa 488-conjugated anti-mouse IgG or Alexa 568-conjugated anti-rabbit IgG (diluted 1:200 in PBS-3% BSA, Life Technologies, USA). Following three additional PBS rinses, the sections were incubated for 5 min at room temperature with 4′,6-diamidino-2-phenylindole (DAPI, Sigma-Aldrich). Normal rabbit IgG was used as a control in the same concentration as the primary antibody. The sections were rinsed as before and mounted on a glycerol-PBS solution (9:1). Single-plane images were obtained using a confocal microscope LSM-META-Zeiss Axioplan 2 (Carl-Zeiss, Germany).

## Electrophoretic mobility shift assays

To generate double-stranded probes, we mixed complementary oligonucleotides in a 1:1 molar ratio, heated them to 95˚C over the course of 5 min, and then gradually cooled them to room temperature. We labeled these double-stranded probes (Table 1) with [γ-$^{32}$P]ATP (Perkin Elmer, USA) using T4 polynucleotide kinase (Invitrogen) or GelRed (Biotium, USA). We performed binding reactions by pre-incubating 50 fmol of each probe, 1 μg of poly(dI-dC), and 5 or 10 μg of nuclear extracts in 1X Binding buffer (20% glycerol, 2.5 mM EDTA, 5 mM $MgCl_2$, 250 mM NaCl, 50 mM Tris-HCl, 2.5 mM DDT) at room temperature. These reactions were then incubated for 30 min at room temperature. For the super-shift assays, 1 μg of rabbit anti-human TAF6, mouse anti-human TAF9, or rabbit anti-*T. solium* TBP1-N was added to the reaction 30 min after nuclear extract; the reaction was then incubated for an additional 30 min at room temperature. The reactions were terminated by adding gel-loading buffer. The complexes were separated on a non-denaturing 5% polyacrylamide gel and visualized using autoradiography (after drying the gel) or in a gel-documentation system.

**Table 1. Double-stranded DNA (dsDNA) probes used for the electrophoretic mobility shift assays.** Downstream promoter element (DPE) or TATA-box information is bolded, and mutated sequence are underlined.

|  | Sequence |
|---|---|
| *Tstbp1*-DPE probe | 5´-TCCGGGTTC**TGTCGA**GCAGATGCAG-3´ |
|  | 3´-AGGCCCAAG**ACAGCT**CGTCTACGTC-5´ |
| Mutated *Tstbp1*-DPE probe | 5´-TCCGGGTTC<u>ATAAAT</u>GCAGATGCAG-3´ |
|  | 3´-AGGCCCAAG<u>TATTTA</u>CGTCTACGTC-5´ |
| Consensus DPE probe | 5´-GAGCCGAGT**GGTTGT**GCCTCCATAG-3´ |
|  | 3´-CTCGGCTCA**CCAACA**CGGAGGTATC-5´ |
| Mutated consensus DPE probe | 5´-GAGCCGAGT<u>ATAAAT</u>GCCTCCATAG-3´ |
|  | 3´-CTCGGCTCA<u>TATATTTA</u>CGGAGGTATC-5´ |
| *Ts2-CysPrx* TATA-box | 5'-GCGCTTCGC**TATATTTG**GCGGTAAG-3' |
|  | 3'-CGCGAAGCG**ATATAAAC**CGCCATTC-5' |
| *Tstbp1* TATA-like 1 (-97 bp) | 5'-ATGTCAACA**TTAAAATT**CTTCCTTG-3' |
|  | 3'-TACAGTTGT**AATTTTAA**GAAGGAAC-5' |
| *Tstbp1* TATA-like 2 (-69 bp) | 3'-ATTTGCCTC**ATTTAAAA**TCCATTTG-5' |
|  | 3'-TAAACGGAG**TAAATTTT**AGGTAAAC-5' |
| Consensus TATA-box | 5'-AAGGGGGGC**TATAAAAG**GGGGTGGG-3' |
|  | 3'-TTCCCCCCG**ATATTTTC**CCCCACCC-5' |

## Homology models of TAF6, TAF9, and DPE probe

The 3D structure model of TsTAF6 (Unitprot A0A1S5YDM4) has retrieved from the extensive AlphaFold high-accuracy protein-structure prediction database (https://alphafold.ebi.ac.uk) [26].

The 3D structure of TsTAF9 was predicted by AlphaFold2 web-based program developed by DeepMind (https://colab.research.google.com/github/sokrypton/ColabFold/blob/main/AlphaFold2.ipynb).

Using the coordinates of the TAF6–TAF9 HFD pair in the human TFIID complex (hTFIID, PDB ID 6MZM) as a template, a rational model of the TsTAF6–TsTAF9 complex was constructed by superposing the previous corresponding modeling structures. The TsTAF6-TsTAF9 complex has subjected to energy minimization in vacuum using the Energy-Minimization module in Molecular Operating Environment (MOE) software (version 2014.09, Chemical Computing Group, Canada).

The DNA-Builder module in MOE was then used to construct the double nucleotide strand of the *Tstbp1*-DPE probe described in Table 1.

## Molecular dynamics simulations

We carried out molecular dynamics simulations using GROMACS version 2020.4 with the AMBER03 force field [27]. The TsTAF6-TsTAF9 complex and six DPE probe molecules, positioned along cartesian axes relatives to the TsTAF6-TsTAF9 complex and placed at a distant of 60 Å from the center of mass of each, were centered in a periodical dodecahedral box. The edges of the box were set 20 Å away from any atom. A total of 144 $Na^+$ and 33 $Cl^-$ ions were added to neutralize the net charge of the system and establish an ionic strength equal to 0.15 mol $L^{-1}$. Next, 98,136 TIP3P [28] water molecules were added to fill the box. The system underwent energy minimization and 100 ns of position-constrained thermal equilibration for all heavy atoms using a harmonic force constant of 1000 kJ $mol^{-1}nm^{-1}$. The molecular dynamics simulations were conducted for 1000 ns using an NPT ensemble at 310.15 K and 1.0 bar for 100 ns. We used the LINCS algorithm [29] to limit the length of all of the covalent bonds. The

time step was 2 fs, and a cutoff of 1.2 nm was applied for short-range electrostatic and Van der Waals interactions; long-range electrostatic forces were treated using the Ewald particle-mesh method [30].

## Results and discussion

### Genomic analysis of *Tstbp1*

The *Tstbp1* genomic sequence spans 1481 bp (Fig 1) and encodes a protein of 238 residues with a predicted molecular mass of 27.6 kDa. This protein contains all the characteristic motifs for TBP, as previously reported [19]. Proximal promoter analysis revealed the presence of putative binding sites for core promoter TFs, such as TBP at -97 bp and -69 bp (TATA-box), TAF6/TAF9 at +27 bp (DPE), TAF1/TAF2 at -2 bp (Inr), YY1 at -119 bp, AP-1 at -57 bp, and NF1 at -192 bp; we also noted two CCAAT boxes at -173 and -167 bp, respectively. The structural coding region comprises five exons separated by four introns (Intron I: 76 bp length, from +192 to +267; Intron II: 208 pb length, from +512 to +719; Intron III: 69 pb length, from +825 to +894; Intron IV: 124 pb length, from +991 to +1114 bp relative to TSS). These introns present the NGT-AGN donor-acceptor splice sites. Each intron also contains a putative U1 recognition sequence (Intron I: GTAAGC; Intron II: GTAAGA; Intron III: GTTTCG; Intron IV: ACAGGT; the donor sites are underlined and a pyrimidine-rich tract for the U2-associated factor (U2AF) binding site (Intron I: TGTCTCTTTTACTTTCAG; Intron II: TTTGCTTCCCTGCTT

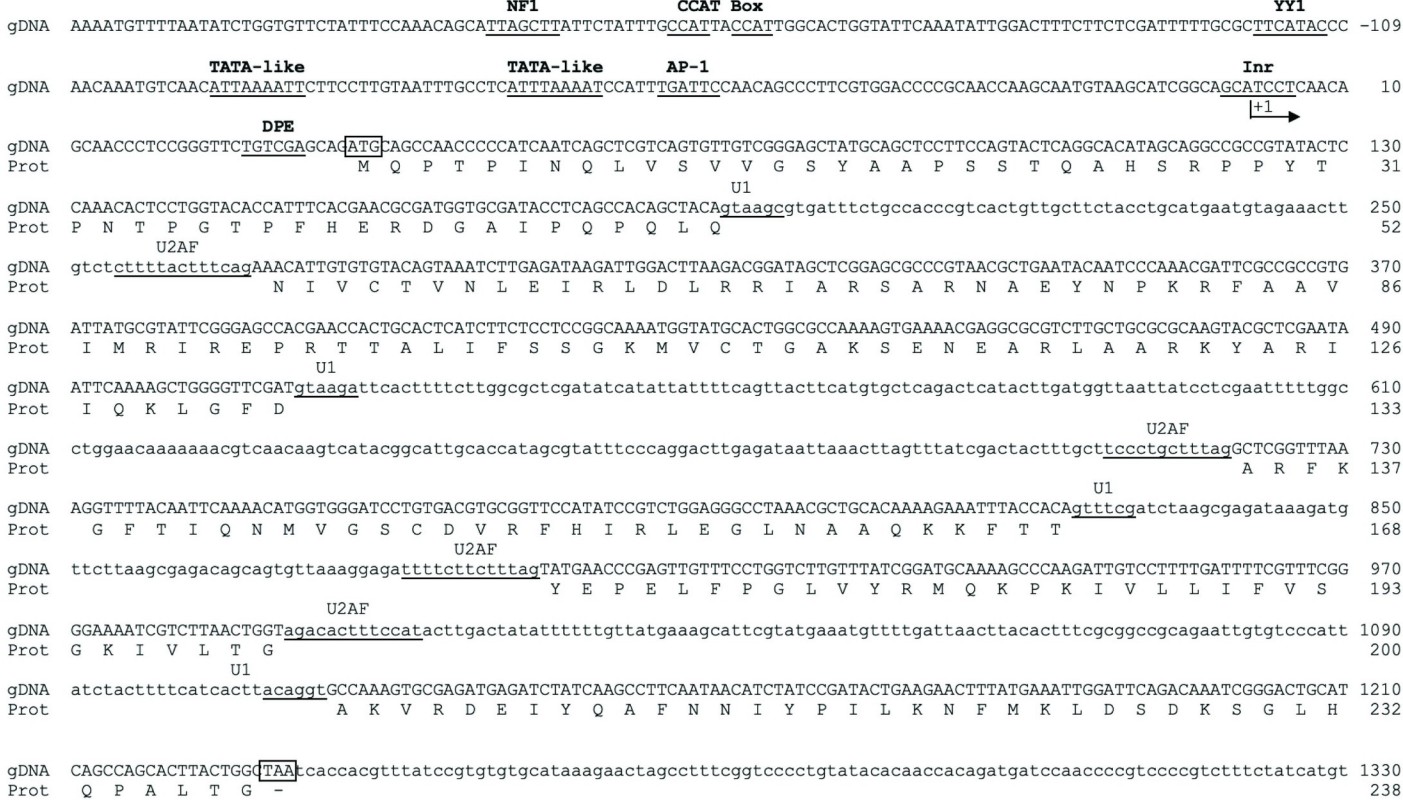

**Fig 1. Genomic structure of *TsTbp1* (GenBank accession number: KX905241).** The Transcription Start Site (TSS) corresponds to an A₊₁ marked with an arrow. Putative TF binding sites indicated above their underlined target sequence. Start (ATG) and Stop (TAA) codons are boxed. Donor and acceptor intron sequences (gt/ ../ag) are bolded. Putative U1 and U2AF splicing binding sites are underlined. Introns are indicated in lowercase text, and numbers to the right correspond to nucleotides and amino acids.

TTA<u>AG</u>; Intron III: GGAGATTTTCTTCTTTA<u>AG</u>; Intron IV: <u>AG</u>ACACTTTCCATACTTG). Those observations reflect the classic structure of a housekeeping gene.

The intron position and gene structure of *TsTbp1* differs from those observed in other organisms, including the *TBP* of human (Fig 2). This situation highlights that structural organization is not relevant for this TF activity. One example of such structure is the region coding for the saddle-shaped that binds DNA. The structure of the N-terminal region varies significantly among species [31–36].

### Transcription start site and gene expression

5′-RACE identified the TSS (Fig 3A) as an adenine ($A_{+1}$) within the initiator (Inr, GC<u>A</u>TCCT), 36 bp upstream of the translation start codon (ATG). This result corroborates previous findings [19]. Southern blot with *T. solium* genomic DNA (Fig 3B) revealed a single-copy gene, evidenced by single bands with *Hind*III and *EcoR*I, and two bands with *BamH*I and *Bgl*II. Additionally, data from the *T. solium* genome project confirm that *Tstbp1* is a single-copy gene. A northern blot (Fig 3C) revealed a primary transcript roughly 1.2 kb in length and a faint band about 1.1 kb in length; these features are likely due to degradation or detection of a homologous gene (e.g., a *Tstrf*). qPCR analysis revealed a relative expression of 1.02 ± 0.092 AU (arbitrary units) in cysticerci and 2.64 ± 0.084 AU in the adult's parasites; these values are

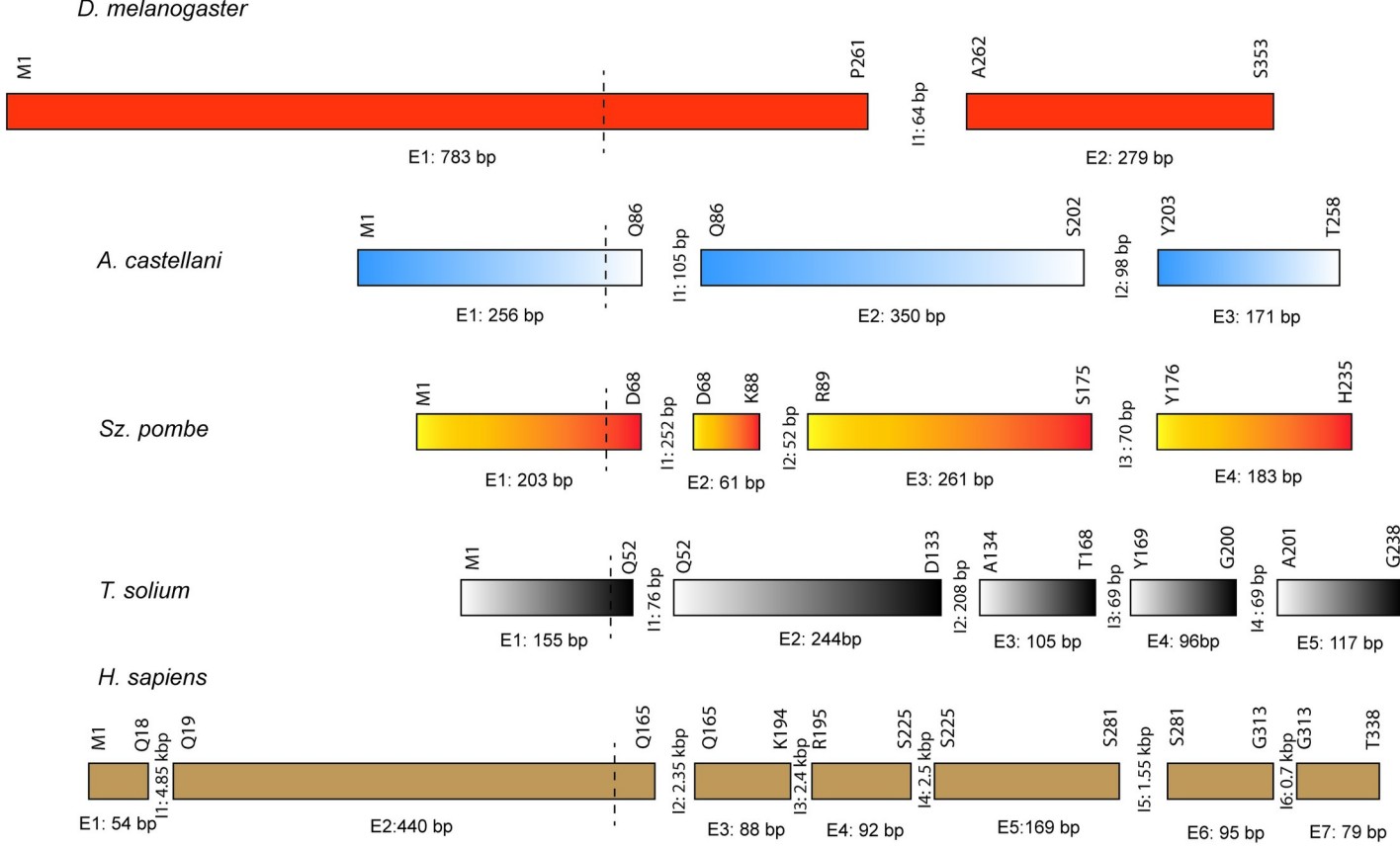

**Fig 2. Gene architecture for the TBPs of different organisms.** The distributions of Exons (E) and Introns (I) in the TBP genes from *D. melanogaster*, *Acanthamoeba castellani*, *Schizosaccharomyces pombe*, *T. solium*, and *H. sapiens* are shown. The number of amino acids within each exon is indicated above the exons, and the size of the exon in bp appears below. The number and size of each intron is indicated between exons. The vertical dashed lines in each gene represent the starting point of the conserved C-terminal domain.

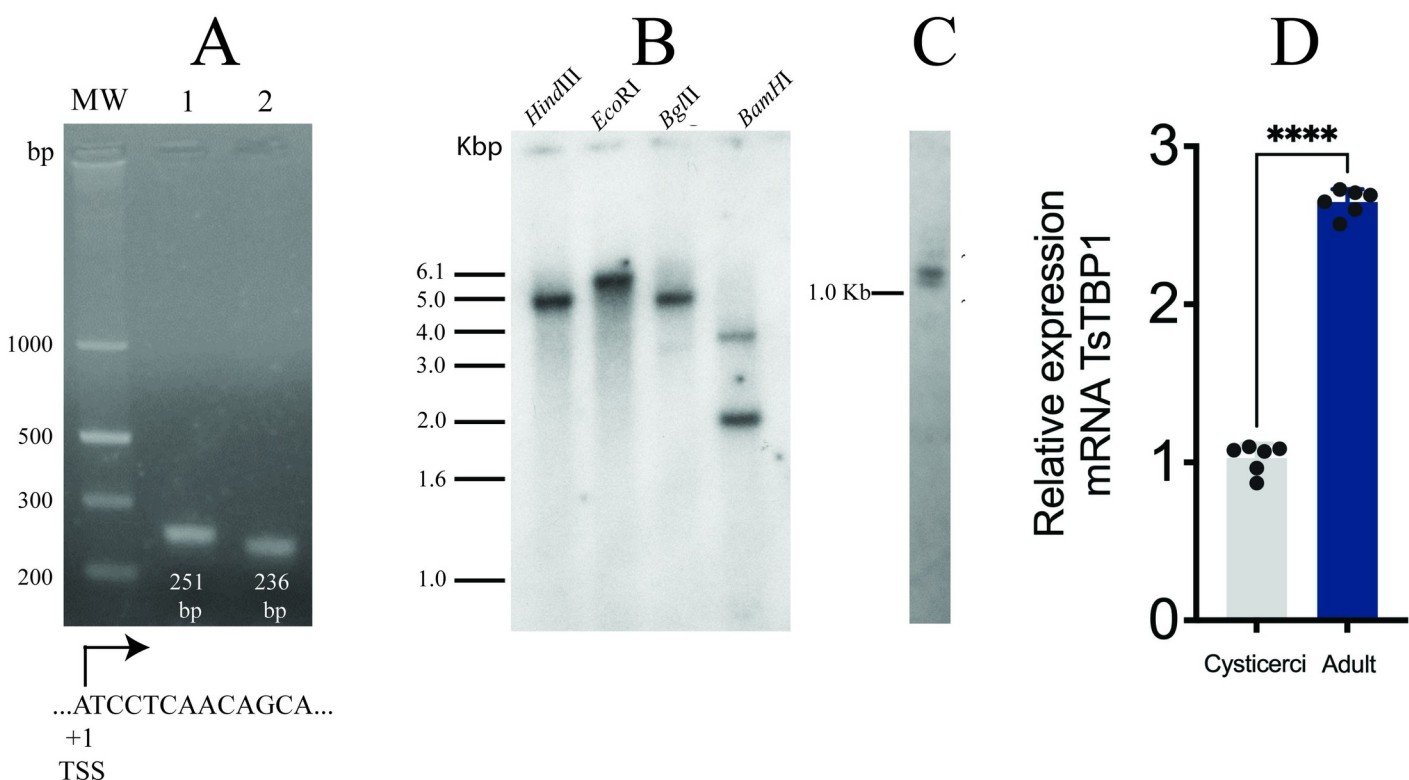

**Fig 3. *Tstbp1* gene analysis.** A) The transcription start site (TSS) was determined by 5´-RACE; agarose gel revealed a 251-bp PCR product (lane 1) and a 236-bp nested-PCR product (lane 2). The gel TSS sequences are shown below. B) Southern blot using gDNA digested with *Hind*III (H3), *Eco*RI (RI), *Bgl*II (B2), and *Bam*HI (B). C) Northern blot of total RNA. Both blots were hybridized with a radiolabeled full-length cDNA-[$^{32}$P] encoding TsTBP1, size markers appear to the left of each blot. D) mRNA relative expression was measured by qPCR for *Tstbp1* in cysticerci and adult stages (n = 6, ****p<0.001, t-test, data are expressed as means ± SEM).

normalized to the *TsCu/ZnSOD* gene (Fig 3D). This finding suggests that TsTBP1 plays a role in growth and egg production in the adult stage; a similar result has been observed in other organisms [37].

## DPE and Inr functionality in the *Tstbp1* core promoter

Given that DPE has never been observed or characterized in *T. solium* promoters, we opted to test its functionality using a luciferase reporter assay. We cloned the core promoter region of *Tstbp1* containing Inr and DPE (WT) into the pGL4.10 vector. Additionally, we also cloned and tested two mutated versions of the core promoter (DPE-less and Inr- and DPE-less) (Fig 4, upper panel). We observed a decrease in luciferase activity of about 30% between the DPE-less and WT constructions. On the other hand, we observed a decrease of about 80% when we compared the Inr- and DPE-less and WT constructions (Fig 4, lower panel). This decrease was similar to that observed in the empty vector control. These results highlight the functionality of the Inr and the DPE located at +27 bp; both elements play important role in the expression of the *Tstbp1* gene. The presence of a DPE in the promoter regions of genes is well-documented in model organisms such as humans and *Drosophila* [7–10]. However, its presence and functionality in the genomes of organisms such as *T. solium* is less understood. The +27 bp position is not common in other organisms that have been studied to date; a +27 bp DPE-like was observed to be functional in the *EphB4* promoter [38].

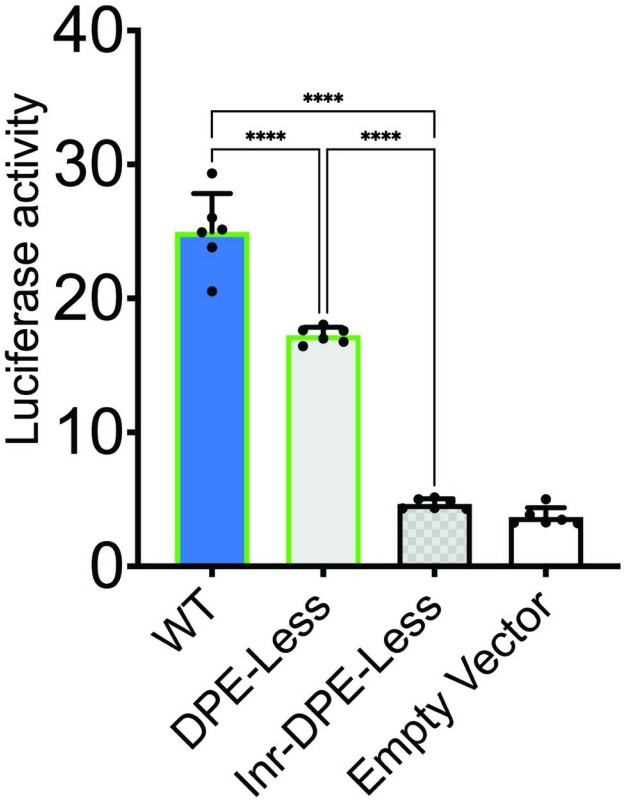

**Fig 4. Functional analysis of the *Tstbp1* core promoter.** Three different *Tstbp1* core promoter constructs were cloned in pGL4.10 vector (upper panel) and used to determine the luciferase activity in HEK-293 cells (lower panel). Cells were co-transfected with each construct and pRL-TK Renilla expression vector. Luciferase activities were measured 24 h post-transfection (n = 6, ****p<0.0001, one-way ANOVA with Bonferroni post hoc test, data expressed as means ± SEM).

### *Tstbp1* is a TATA-less promoter

Electrophoretic mobility shift assay (EMSA) analysis of nuclear extracts and the TATA-like elements -97 and -69 bp relative to the TSS (Fig 5) revealed no shifted bands. That finding is at odds with the binding observed for the *Ts2-cysPRX* TATA-box located -30 bp relative to the TSS [19]. This result reveals that *Tstbp1* is a TATA-less gene, consistent with the pattern observed in many housekeeping genes. Interestingly, the presence of DPE and TATA-box is not typically observed in the same core promoter, but they are present in the core promoters of developmental-related genes [39].

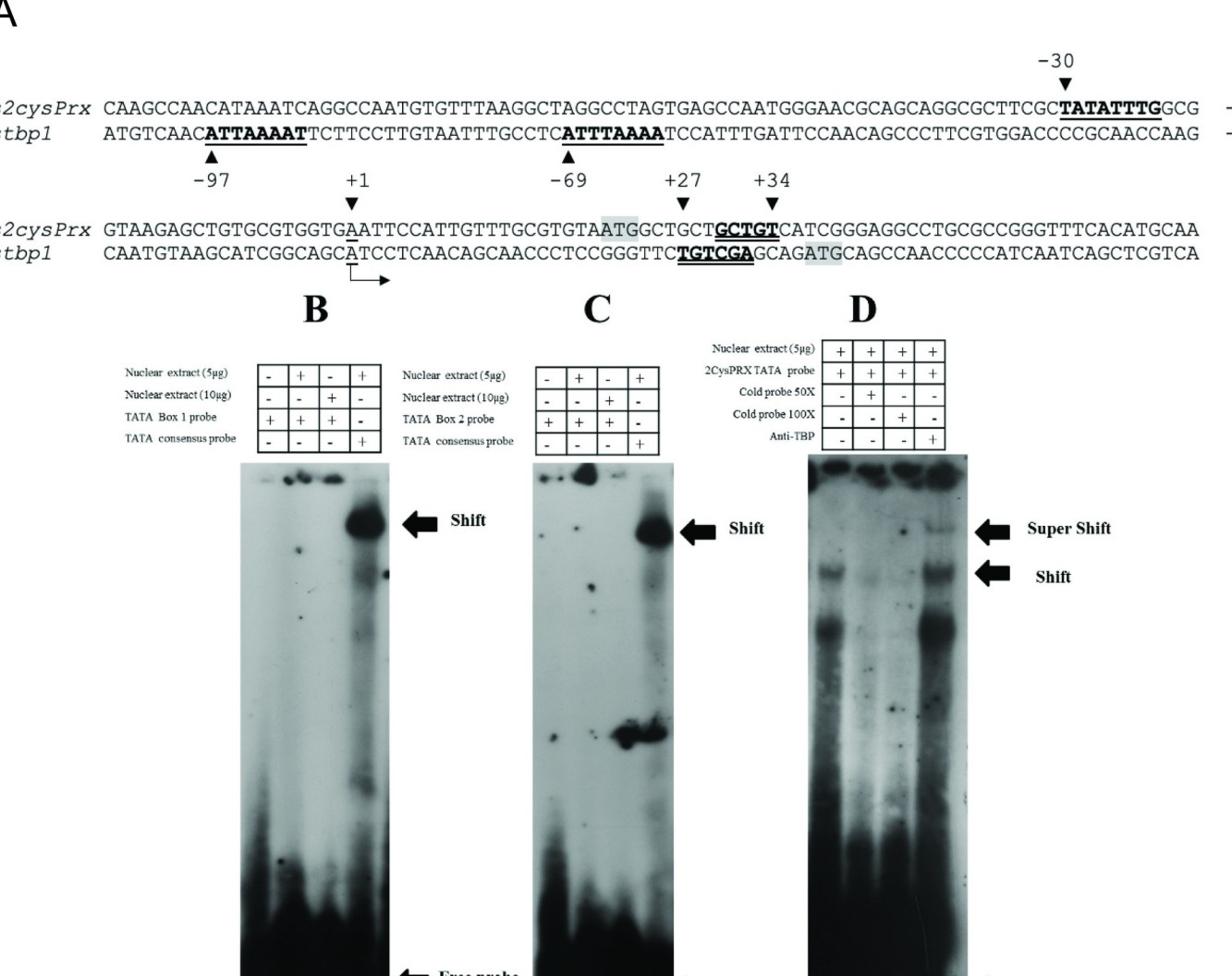

**Fig 5. EMSAs testing TATA-like elements present in *Tstbp1* promoter.** A) Alignment of the *Tstbp1* and *Ts2-cysPrx* promoters revealed TATA-like elements at -97 and -69 bp and the TATA-box at -30 bp (underlined). Putative DPE for each gene over the +27 to +34 bp region is indicated with a double underline. The TSS is indicated with an arrow. B) The EMSA for the probe containing TATA-like element at -97 bp in the *Tstbp1* gene. C) The EMSA for the probe containing TATA-like element at -69 bp in the *Tstbp1* gene. D) The EMSA and super-shift assay for the probe containing the TATA-box of the *Ts2-cysPrx* gene and the anti-*T. solium* TBP1-N antibody.

## Identification of TsTAF6 and TsTAF9 in *Taenia solium*

When we detected a functional DPE in the *Tstbp1* promoter, we opted to investigate factors interacting with this element. Mining the WormBase Parasite genome project database revealed the presence of TsTAF6 and TsTAF9, which are both known to interact with DNA [8–10]. PCR amplification confirmed the cDNA encoding TsTAF6 and TsTAF9. The resulting sequences revealed that TsTAF9 contains 242 amino acids and has a predictive pI of 4.8 and molecular weight of 25.98 kDa (GenBank: PP763292, S1 Fig). On the other hand, TsTAF6 is composed of 645 amino acids and has a predictive pI of 9.4 and a molecular weight of 70.42 kDa (GenBank: KY124274.1, S2 Fig). Due to the lack of specific antibodies for TAF6 and

TAF9 of *T. solium*, commercial antibodies that recognize human TAFs were used. An analysis of their antigenic regions showed homologous epitopes in both species on the region used for the preparation of the antibodies (S3 Fig). By western blot we identify TBP and TsTBP1 with anti-human TBP and anti-TsTBP1 antibodies, respectively. Likewise, TAF9 and TAF6 in nuclear extracts from HEK-293 cells and its homologs in *T. solium* cysticerci with the anti-human TAF6 and TAF9 antibodies (Fig 6A). Confocal microscopy using DAPI, anti-TAF6, and anti-TAF9 antibodies revealed the presence of TAF6 and TAF9 in the nucleus of cells that constitute the bladder wall in *T. solium* cysticerci. We found that TsTAF6 and TsTAF9 are highly localized inside the nucleus and close to the nuclear membrane. On the other hand, significant but incomplete overlap was observed in the spatial distributions of the factors. That finding is suggestive that functional transcription occurred in some places (Fig 6B). However, there are likely other locales in which transcription did not occur or incomplete transcription initiation complexes, inhibitory complexes, or storage sites persisted instead.

## Interaction of TsTAF6 and TsTAF9 with the DPE of the *TsTbp1* promoter

Fig 7 shows EMSA results of the interactions between *TsTbp1* and DPE. In the assay, we observed: lane 1, the free run control, with no bands; lane 2 to 4, two shifted bands are observed that decreased when different amounts of nuclear extracts are incubated with *TsTbp1*-DPE probe (Table 1); lanes 5 and 6, display a super-shifted band when anti-TAF6 and anti-TAF9 antibodies are added to the reaction, respectively; lane 7, shows the shifted bands produced by incubating nuclear extracts, and the consensus *D. melanogaster* DPE-probe (DmDPE); finally, in lanes 8 and 9, the disappearance of any band when using DmDPE mutated and *TsTbp1*-DPE mutated. We identified a TGTCG motif located at +27 to +31 bp

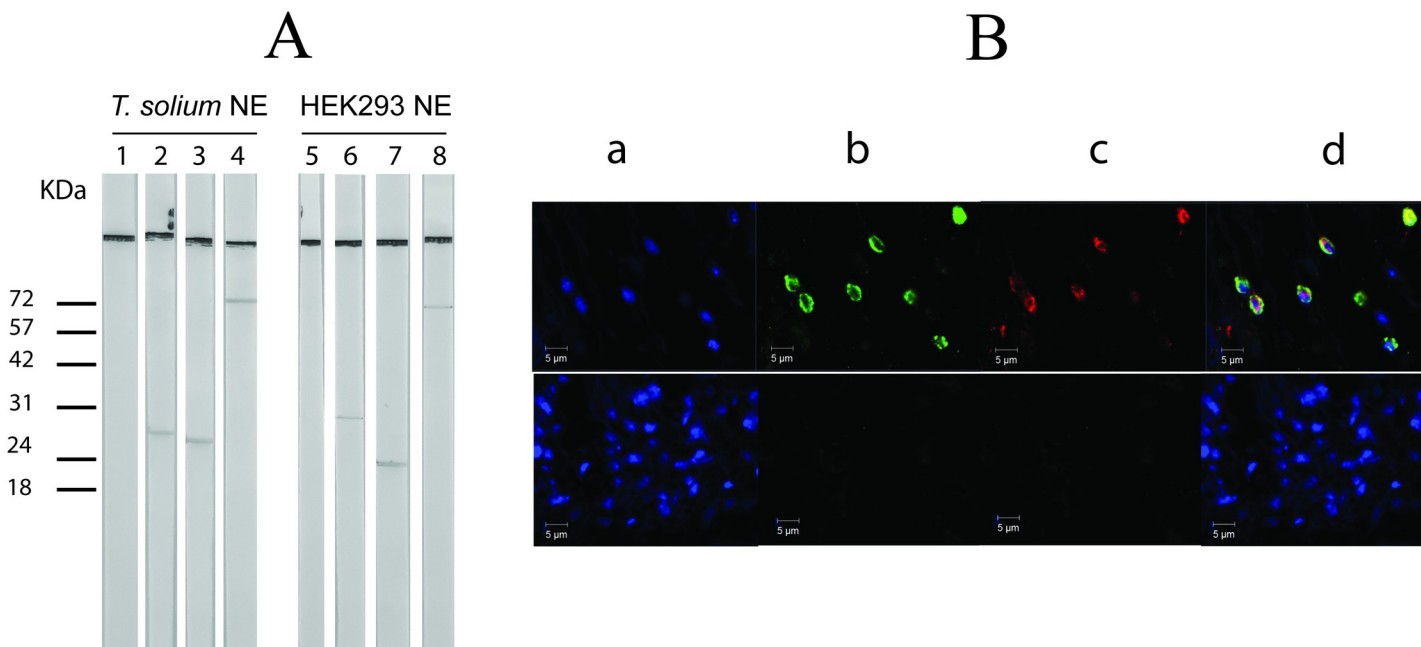

**Fig 6. Immunodetection of TAF6 and TAF9 by western blot.** A) Nuclear extracts of *T. solium* (lanes 1–4) and HEK-293 cell (lanes 5–8) were confronted with a normal rabbit IgG (lanes 1 and 5), anti-*T. solium* TBP1-N (lane 2), anti-human TBP (lane 6), anti-human TAF9 (lanes 3 and 7) and anti-human TAF6 (lanes 4 and 8). B) Sections of the bladder walls of cysticerci were incubated with DAPI, anti-human TAF9, and anti-human TAF6 (upper panel) or DAPI and normal rabbit IgG (lower panel). Followed by Alexa 488-conjugated anti-mouse IgG and Alexa 568-conjugated anti-rabbit IgG. The sections were observed in a) blue channel (DAPI), b) green channel (Alexa 488), c) red channel (Alexa 568), and d) merged signals.

| Nuclear extract | 0µg | 10µg | 5µg | 2.5µg | 10µg | 10µg | 10µg | 10µg | 10µg |
|---|---|---|---|---|---|---|---|---|---|
| *Tstbp1*-DPE probe | + | + | + | + | + | + | - | - | - |
| Anti-TAF6 antibody | - | - | - | - | + | - | - | - | - |
| Anti-TAF9 antibody | - | - | - | - | - | + | - | - | - |
| Consensus DPE | - | - | - | - | - | - | + | - | - |
| Consensus DPE mut | - | - | - | - | - | - | - | + | - |
| *Tstbp1*-DPEmut probe | - | - | - | - | - | - | - | - | + |

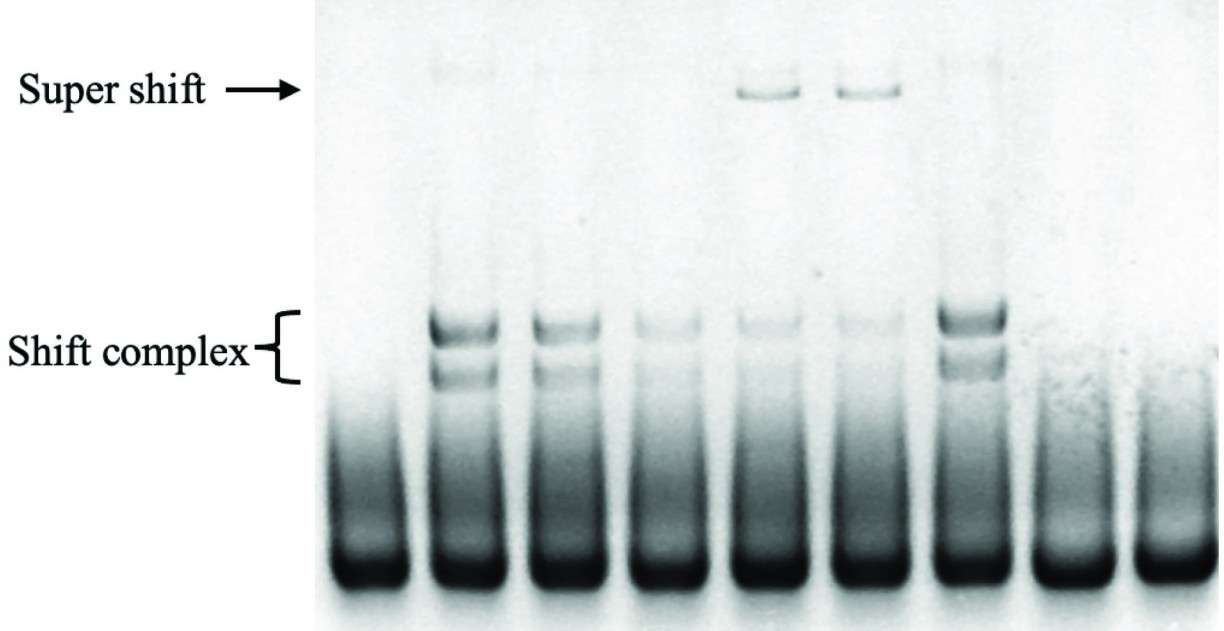

**Fig 7. The interaction of TsTAF6 and TsTAF9 with *TsTbp1*-DPE from an EMSA.** Lane 1: Labeled dsDNA without nuclear extract; lanes 2, 3, and 4: Different quantities of nuclear extract interaction with the *TsTbp1*-DPE-probe; lane 5: The super-shift interaction using anti-human TAF6 antibody; lane 6: The super-shift interaction using anti-human TAF9 antibody; lane 7: Consensus DPE-probe interaction with nuclear extract (positive control); lane 8: Mutated consensus DPE-probe interaction with nuclear extract (negative control); lane 9: Mutated *TsTbp1*-DPE-probe interaction with nuclear extract. See Table 1 for the probe sequences.

that was strongly similar to consensus DPE (A/G-G-A/T-C/T-G/A/C); this motif is known to bind TAF6 and TAF9 [7–10]. The DPE, typically present in promoters lacking TATA-box (e.g., *TsTbp1*), is indicative of an interaction between TsTAF6 and TsTAF9 with the DPE element; these factors might also possibly bind to other promoters of *T. solium* genes. However, follow-up research is necessary to determine the functionality of this element in other *T. solium* promoters.

## TsTAF6 and TsTAF9 molecular modeling and molecular dynamics

To corroborate the interaction between TsTAF6 and TsTAF9 and the DPE observed in the EMSA, we modeled the TsTAF6-TsTAF9-DPE complex. Recent work has used cryo-EM, chemical cross-linking mass spectrometry, and biochemical reconstitution to determine the structure of hTFIID. hTFIID consists of flexible trilobed complexes (lobes A, B, and C) composed of the TBP and 13 TAFs (TAF1–TAF13) [13,40]. The cores of lobes A and B are

composed of HFD heterodimers of TAF6-TAF9, TAF4-TAF12, and TAF8-TAF10; that finding supports previous studies [41,42]. Shao et al. proposed that the HFD pair in the TAF6-TAF9 dimer may be involved in interactions with the DNA promoter; although this dimer in hTFIID is located quite far from the DNA binding site [10]. Although the regulation of the transcription initiation mechanism is likely subtly different in mammalians and cestodes, TFIID in any case is responsible for initially recognizing the core promoter.

We used the coordinates of the TAF6-TAF9 HFD pair in hTFIID (PDB-ID: 6MZM) as a template to construct a homologous rational model of TsTAF6-TsTAF9. The model of TsTAF6 (645 amino acids, MM 70,42 kDa, S1 Fig) retrieved from the AlphaFold protein structures database strongly resembles TAF6 in the hTFIID complex in structure. In this model, the HEAT repeat domain and the N-terminal HFD are separated by a flexible, unstructured segment. In the AlphaFold model of the TsTAF9 (242 amino acids, MM 25,97 kDa, S2 Fig), the HFD domain superposes properly with the corresponding HFD in TAF9 in hTFIID. The final model of the TsTAF6-TsTAF9 dimer assembled via superposition is shown in Fig 8A.

Analysis of the electrostatic potential surface of this modeled complex revealed a prominent positively charged patch in the TsTAF6-TsTAF9 HFD pair (Fig 8B). This finding suggests that this region has a specific electrostatic function (e.g., a DNA-binding site). Similarly, there is a positive patch in the histone octamer in the nucleosome core particle of chromatin bound to DNA (Fig 8C; PDB-ID 1AOI) [43]. We accordingly explored the structure association between the DPE-probe (Table 1) and the modelled TsTAF6-TsTAF9 dimmer using molecular dynamics simulations (Materials and Methods).

The DPE probe initially placed in front of the positive patch in the TsTAF6-TsTAF9 HDF pair bound to form an incipient TsTAF6-TsTAF9-DPE probe complex after 300 ns of simulation. Subsequent conformational changes led to an increase in the contact area between the TAF molecules and the DPE probe. The complex after both 300 and 1000 ns of simulation is

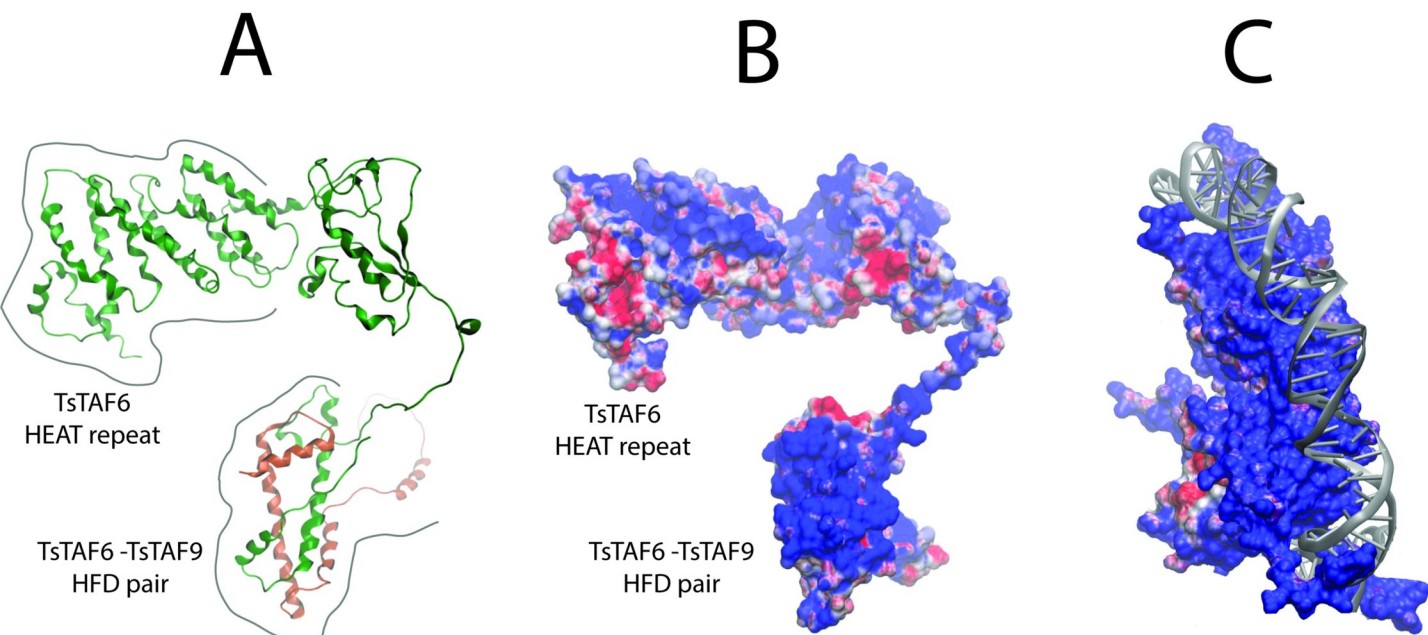

**Fig 8. Molecular modeling of the TsTAF6-TsTAF9 complex.** A) Optimal model of the TsTAF6-TsTAF9 dimer assembled via superposition using the TAF6-TAF9 pair in human TFIID as a template (PDB-ID 6MZM). B) Electrostatic potential surface of the TsTAF6-TsTAF9 complex showing prominent positive patches in the HFD (blue). C) The positive patch in the histone octamer in the nucleosome core particle of chromatin bound to DNA (PDB-ID 1AOI).

shown in Fig 9. The DPE probe and TsTAF6-TsTAF9 HFD interacted via electrostatic and van der Waals forces and hydrogen bonds after 1000 ns of stimulation. In particular, ARG6, LYS7, LYS8 ASN10, ARG11, and LYS15 from TsTAF6 and ASN18, PRO19, SER13, VAL14, SER16, VAL17, SER20, and ASP29 from TsTAF9 directly interacted with the 5´-TGTCGA-3´/3´-ACAGCT-5´ sequence. The result was an extensive network of hydrogen bonds and salt-bridge interactions. Although we did not study the evolution of the complex for stimulation durations exceeding 1 microsecond, we noted that the interaction between the participating molecules initiated at 300 ns remained stable for 700 ns. That finding implies a strong union between TsTAF6-TsTAF9 and DPE probe.

## DPE in gene transcription

Downstream promoter element was first identified in *Drosophila* spp TATA-less genes. In these genes, purified TFIID protects the +1 to +35 bp region from DNAse I degradation. There is no protection from DNAse I in the +25 to +30 bp region in TATA-containing promoters, however. The A/G-G-A/T-C-G-T-G sequence, located around +30 bp relative to the TSS, has been determined to be a conserved motif [7]. Photo-cross linking experiments with purified TFIID have

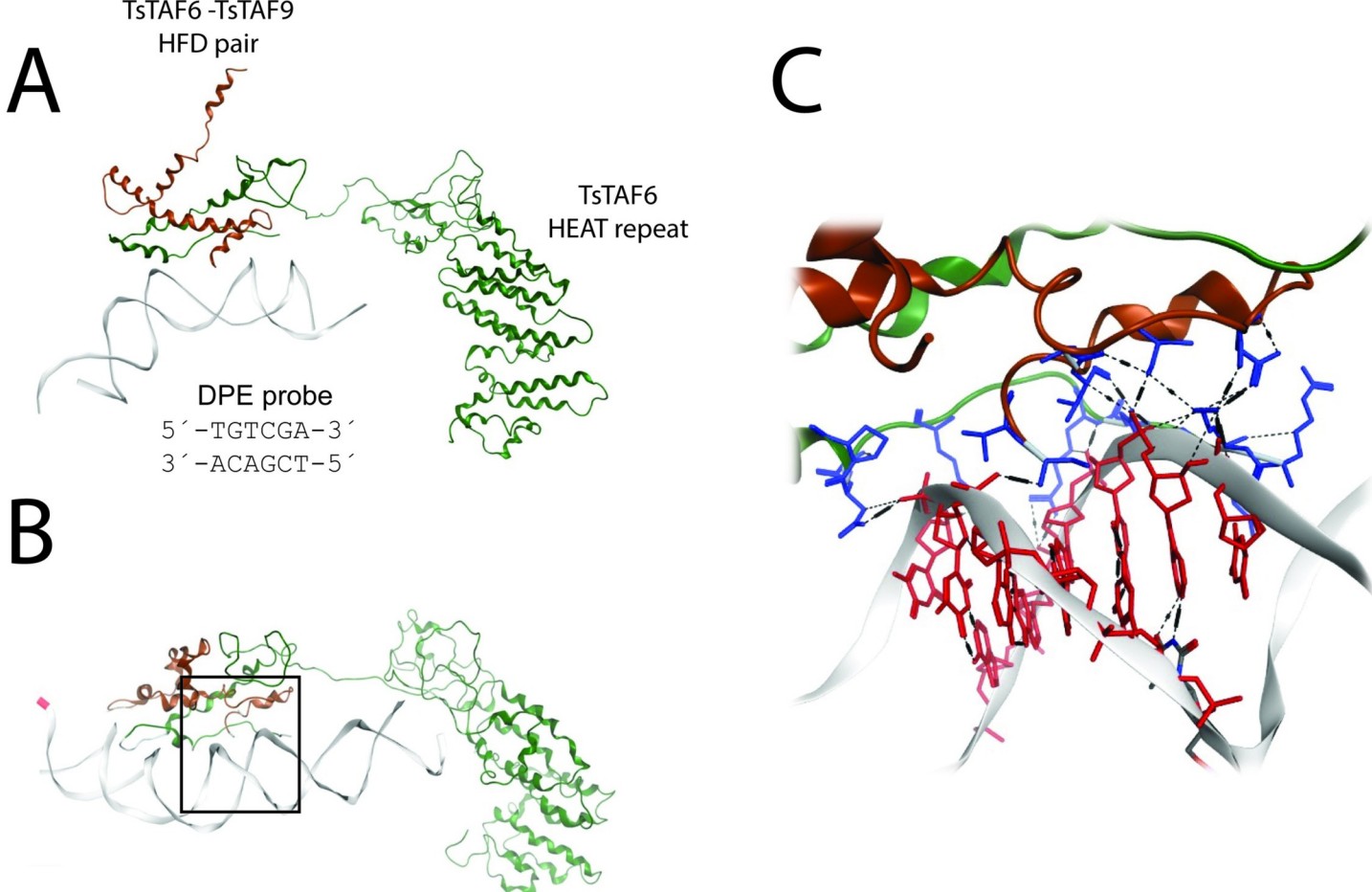

**Fig 9. Model of the TsTAF6-TsTAF9-DPE probe complex obtained from molecular dynamics simulations.** A) Incipient TsTAF6-TsTAF9-DPE probe complex formed after 300 ns of simulation. B) TsTAF6-TsTAF9-DPE probe complex formed after 1000 ns of simulation. C) Close-up of the interaction region between the DPE probe and the amino acid residues of the HFD pair in the TsTAF6-TsTAF9 complex.

demonstrated a specific interaction of dTAF$_{II}$60 and dTAF$_{II}$40 (now known as TAF6 and TAF9, respectively [44]) with a TATA-less and DPE-containing promoter, in a likely heterotetramer conformation [9]. These interactions also extend to the ten-element motif [45,46]. The functionality of DPE requires a specific interspace distance and is enhanced by the presence of a G in the +24 bp position. An analysis of 205 *Drosophila* promoters found that 43% contained a TATA-box, 40% possessed a DPE, and 31% had neither [5]. Interestingly, factors such as Mot1 and NC2 positively influence transcription regulation. However, TAF4 depletion using RNA interference (RNAi) appears to decrease transcription rates in both types of promoters [47]. Now it is known that TAF4, as part of the TAF4-TAF12 complex, does bind to DNA. The complex has been shown to bind DNA with high affinity, likely via one TAF4-TAF12 heterodimer forming several contacts with a single DNA molecule [48].

Recent cryo-EM work has revealed that a TAF6 and TAF9 in a heterodimer structure formed an HFD in the lobe B of the PIC. However, neither TAF6 nor TAF9 were close enough to contact the DPE from the SCP [13,40]. Moreover, the *HDM2* (TATA- and DPE-containing) and *PUMA* (TATA-less and DPE-containing) promoters exhibited a similar PIC conformation compared with that of the SCP [18].

It is possible that differences in isosurface potential between our computed TAF6-TAF9 complex and the human TFIID complex are responsible for the increased function in cestodes and related organisms (e.g., *Plasmodium falciparum* and *Trypanosoma cruzi*) [49–51]. Such studies can lead to a specific or general interaction with the DPE probe; this hypothesis can be tested with molecular dynamics simulations. However, there are currently no three-dimensional models for TFIID for cestodes; we accordingly cannot compare our results. Nevertheless, we propose the possibility of a direct interaction between both, TAF6 and TAF9, and the DPE in *T. solium* promoters.

## Conclusions

We analyzed the *Tstbp1* gene, which is characterized by its housekeeping gene structure and includes key motifs such as a functional Inr and a DPE at the unusual position of +27 bp. This gene also includes binding sites for some TFs in its core promoter. The specific arrangement of exons and introns in *Tstbp1* is similar to that of related organisms but differs significantly from the arrangement observed in humans and other higher organisms. Moreover, our EMSA, molecular modeling, and molecular dynamics simulations suggest a mechanism of interaction between the TsTAF6-TsTAF9 complex and the DPE of *Tstbp1* via HFD; the HFD in our complex exhibits a positive isosurface potential that is absent in the human TAF6-TAF9 complex. However, it is possible that the TAF6-TAF9 complex could bind via an indirect interaction mediated by TAF4 or another TF, as has been observed with cryo-EM in recent years. Advancing our knowledge of cestode transcriptional systems is not only of academic interest but also of practical importance for understanding the biology of these parasites.

## Supporting information

**S1 Fig. TsTAF9 cDNA and protein sequence obtained by PCR amplification from a cDNA *T. solium* cyst library.** Numbers to the right corresponds to nucleotide or amino acids respectively (GenBank: PP763292).
(PDF)

**S2 Fig. TsTAF6 cDNA and protein sequence obtained by PCR amplification from a cDNA T. solium cyst library.** Numbers to the right corresponds to nucleotide or amino acids

respectively (GenBank: KY124274.1).
(PDF)

**S3 Fig.** Alignment comparison for amino acid sequence A) TAF9 from Homo sapiens (HsTAF9) and T. solium (TsTAF9, from GenBank: PP763292). B) TAF6 from Homo sapiens (HsTAF6) and T. solium (TsTAF6, GenBank: KY124274.1). In yellow is highlighted the region used as immunogen for antibodies development according to manufacturer. Underlined amino acids represents the epitopes for both species, predicted using BCPREDS web server (http://ailab-projects2.ist.psu.edu/bcpred/predict.html). Asterisks (*) denotes identical amino acids and colon (:) denotes homologous amino acids.
(PDF)

**S1 Raw images.**
(PDF)

## Acknowledgments

We thank the Dirección de Cómputo y de Tecnologías de Información y Comunicación (Miztli) and the Laboratorio de Supercómputo y Visualización en Paralelo at the Universidad Autónoma Metropolitana-Iztapalapa for access to their computer facilities. Also to Dirección General de Asuntos del Personal Académico (DGAPA, PAPIIT), UNAM. Thanks to MSc Alicia Ochoa-Sánchez for technical help.

## Author Contributions

**Conceptualization:** Oscar Rodríguez-Lima, Ponciano García-Gutiérrez, Abraham Landa.

**Data curation:** Oscar Rodríguez-Lima, Ponciano García-Gutiérrez, Lucía Jiménez, Angel Zarain-Herzberg, Roberto Lazzarini, Karel Estrada, Abraham Landa.

**Formal analysis:** Oscar Rodríguez-Lima, Ponciano García-Gutiérrez, Angel Zarain-Herzberg, Karel Estrada, Abraham Landa.

**Funding acquisition:** Oscar Rodríguez-Lima, Abraham Landa.

**Investigation:** Oscar Rodríguez-Lima, Ponciano García-Gutiérrez, Lucía Jiménez, Laura A. Velázquez-Villegas, Angel Zarain-Herzberg, Roberto Lazzarini, Karel Estrada, Abraham Landa.

**Methodology:** Oscar Rodríguez-Lima, Ponciano García-Gutiérrez, Lucía Jiménez, Laura A. Velázquez-Villegas, Angel Zarain-Herzberg, Roberto Lazzarini, Abraham Landa.

**Project administration:** Oscar Rodríguez-Lima, Abraham Landa.

**Resources:** Oscar Rodríguez-Lima, Abraham Landa.

**Software:** Oscar Rodríguez-Lima, Ponciano García-Gutiérrez, Karel Estrada, Abraham Landa.

**Supervision:** Oscar Rodríguez-Lima, Abraham Landa.

**Validation:** Oscar Rodríguez-Lima, Lucía Jiménez, Angel Zarain-Herzberg, Roberto Lazzarini, Abraham Landa.

**Visualization:** Oscar Rodríguez-Lima, Abraham Landa.

**Writing – original draft:** Oscar Rodríguez-Lima, Ponciano García-Gutiérrez, Abraham Landa.

**Writing – review & editing:** Oscar Rodríguez-Lima, Ponciano García-Gutiérrez, Lucía Jiménez, Angel Zarain-Herzberg, Roberto Lazzarini, Karel Estrada, Abraham Landa.

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
