## [Decision Letter · Decision Letter 0]

2 Feb 2024

PONE-D-23-43961Taenia solium TAF6 and TAF9 bind to a downstream promoter element present in the Tstbp1 gene core promoterPLOS ONE

Dear Dr. Landa,

Thank you for submitting your manuscript to PLOS ONE. After careful consideration, we feel that it has merit but does not fully meet PLOS ONE’s publication criteria as it currently stands. Therefore, we invite you to submit a revised version of the manuscript that addresses the points raised during the review process.

As detailed in their evaluations, all three expert reviewers have raised substantial concerns with regards to the experiments, necessary controls and interpretation. Please address all major concerns of the reviewers. To improve the manuscript, I recommend addressing as many minor comments as possible.

We look forward to receiving your revised manuscript.

Kind regards,

Tamar Juven-Gershon, Ph.D.

Academic Editor

PLOS ONE

3. To comply with PLOS ONE submissions requirements, in your Methods section, please provide additional information regarding the experiments involving animals and ensure you have included details on (1) methods of sacrifice, (2) methods of anesthesia and/or analgesia, and (3) efforts to alleviate suffering.

Reviewers' comments:

Reviewer's Responses to Questions

**Comments to the Author**

1. Is the manuscript technically sound, and do the data support the conclusions?

Reviewer #1: No

Reviewer #2: Partly

Reviewer #3: Partly

2. Has the statistical analysis been performed appropriately and rigorously? 

Reviewer #1: I Don't Know

Reviewer #2: N/A

Reviewer #3: No

3. Have the authors made all data underlying the findings in their manuscript fully available?

Reviewer #1: Yes

Reviewer #2: Yes

Reviewer #3: Yes

4. Is the manuscript presented in an intelligible fashion and written in standard English?

Reviewer #1: Yes

Reviewer #2: Yes

Reviewer #3: Yes

5. Review Comments to the Author

Reviewer #1: In this manuscript, the authors examined the promoter of the TBP gene (Tstbp1) in Taenia solium. By using 5'-RACE with total RNA from T. solium cysticerci, they identified the transcription start site (TSS) (Fig. 3A). Examination of the promoter sequence suggested the presence of a DPE motif at the unconventional +27 position relative to the A+1 (Fig. 1). The Tstbp1 promoter appears to lack a TATA box at the normal position (roughly -30 relative to the TSS), but there are TATA-like sequences at positions -97 and -69 relative to the TSS. Analysis of these TATA-like sequences by gel shift experiments with T. solium nuclear extracts did not reveal any DNA-binding activity (Fig. 4).

Because of the existence of a potential DPE in the Tstbp1 promoter, the authors investigated TAF6 and TAF9 in greater detail, as it had been previously found that TAF6 and TAF9 can crosslink and potentially bind to the DPE. Western blot analysis of T. solium nuclear extracts with anti-TsTBP1, anti-human TAF6, and anti-human TAF9 revealed bands that appear to correspond to the cognate proteins. Immunofluorescence of cysticerci with the anti-human TAF6 and TAF9 antibodies yielded images that may correspond to the cognate T. solium TAF6 and TAF9. Gel shift analyses with T. solium nuclear extracts and wild-type or mutant putative DPE sequences yielded bands that gave an apparent supershift upon addition of anti-human TAF6 as well as anti-human TAF9 (Fig. 6). Molecular modeling of TsTAF6 and TsTAF9 was also performed (Figs. 7 and 8), and suggested that TsTAF6-TsTAF9 dimers might interact with the putative DPE.

The authors' efforts to analyze the core promoter of the TBP gene in a challenging and important organism are appreciated. There are, however, too many unknowns to be able to recommend this manuscript for publication in PLoS ONE. Some of the issues are as follows.

1. First, it is not known that the putative DPE-like motif in the promoter is a functionally active DPE. Some sort of transcription assay with wild-type versus mutant DPE versions of the promoter should be performed to assess whether or not the DPE-like motif is important for core promoter activity. In addition, it is questionable whether the sequence is a genuine functional DPE, as it is located at the +27 position rather than the standard +28 position. Most importantly, it must be determined whether or not the putative DPE is a functionally active DPE.

2. The gel shift analysis in Fig. 6 does not provide any evidence whether TAF6 and/or TAF9 interact directly with the DPE sequence. The shifted bands seen with the nuclear extract could be due to the binding of non-TAF6/9 subunits of a multiprotein complex (such as TFIID) that also contains TAF6 and TAF9. In that case, the antibodies against TAF6 and TAF9 would cause a supershift. It might be possible to clarify this point by carrying out the gel shift analyses with purified TsTAF6 and TsTAF9 proteins.

Reviewer #2: The work of Rodriguez-Lima et al. aims to expand our understanding of the transcription machinery in cestodes. TFIID is a general transcription factor found across species that consists of the TATA binding protein (TBP) and a set of highly conserved TBP associated factors (TAFs). The Landa lab previously cloned the cDNA for the Taenia solium TATA binding protein (TsTBP1).This manuscript builds on that work by characterizing the TsTBP1 gene structure, regulatory elements and expression levels using standard techniques.

The author's more significant contribution is the identification and characterization of two addition TFIID subunit in Taenia solium, TAF6 and TAF9. Antibodies against human TAF6 and TAF9 proteins were used in westerns blots to identify proteins of the predicted molecular weight based on the cDNA sequence. The antibodies also were used in confocal microscopy to localize the proteins to the nucleus. What is the evidence that the antibodies against human TAF6 and TAF9 antibodies will recognize the Taenia solium proteins in western blots? A figure that illustrates the degree amino acid sequence homology between the human and Taenia solium protein would be helpful. The authors might consider expressing TsTAF6 and TsTAF9 and verify the cross-reactivity of the antibodies being used for these studies.

TAF6 and TAF9 have been shown to bind to the downstream promoter element (DPE), which was first discovered downstream of transcription start sites in drosophila. The authors set out to investigate if this also is the case in Taenia solium using EMSA, homologous competition and human TAF6 and TAF9 antibodies. As shown in Figure 6, mobility shifted complexes are detected and decrease in intensity with the addition of unlabeled competitor probe. Once again their is concern about the specificity of antibodies being used for the supershift experiments. The appearance of the supershifted complex due to the binding of the TAF6 or TAF9 antibody is not accompanied by any detectable decrease in the TAF6- or TAF9-DNA complexes.

The last sentence in the abstract states "Novel and interesting features of the TsTAF6-TsTAF9 complex for interaction with DPE on T. solium promoters are discussed." The modeling of TAF6-TAF9 dimer binding to DNA is a powerful approach and great starting point. However, it was not clear what new information was uncovered that increased our understanding about the interaction of TAF6 and TAF9 with promoter DNA.

Reviewer #3: PONE-D-23-43961

In this manuscript, Rodriguez-Lima et al. reports the characterization of the Taenia solium cestode TBP gene (TsTbp1) and of its promoter which contains 2 TATA-like elements, an Inr and one DPE element. They showed that the TATA-like elements are not bound by TsTBP1. They cloned the genes encoding TsTAF6 and TsTAF9 and reported an interaction between these proteins and TsTbpP1 DPE element by EMSA. They build a model of the interaction between TsTAF6, TsTAF9 and DPE DNA.

I have some concerns that the data are interesting but the potential novelty is not really highlighted. There are already lots of data published on Pol II transcription initiation and the cloning and characterization of the TAFs in yeast, insects or mammals were carried out several decades ago. The authors should explain more what their new cestode model brings to the Pol II transcription initiation field.

Figure 2: The authors compare the genomic structure of the Tbp gene in different species. I do not understand the interest of this figure and I am not convinced by their conclusion that “the intron position and gene structure in TsTbp1 are similar to those observed in human Tbp1, highlighting the importance of this transcription factor across different organisms”: the human Tbp gene has 7 exons, TsTbpl1 only 5, the size of the exons is not similar, the position of the introns is not similar… To me, the real conclusion of this figure is that there is a huge discrepancy in the organization of these orthologs…

Figure 3B: It is not clear why the intensity of the upper band is weaker than the lower band. Is the BglII site located close to the extremity of the region recognized by the probe? It would be informative to have a map of the locus with the restriction sites (at least for BglII). The conclusion of the authors is that TsTbp1 is present in a single copy. This need to be clarified: does this mean that there is no Tbp paralog? As Trf2/Tbpl1 is present in metazoans, including C elegans, if true, this would be an interesting observation. The hybridization conditions are not described in the material and methods and it is not possible to know whether they are stringent or not.

In Figure 3D, the authors used SOD6 as a normalizing gene for the RT-qPCR performed on cysticerci and adults. This is not a classical housekeeping gene used (at least in other models…): what is the evidence that it is a good marker? Are the data comparable when using Gapdh or 18S RNA?

Line 267: from the RT-qPCR experiment, the authors conclude that TsTBP1 might be involved in growth and egg production in the adult stage, citing Persengiev (1996) Mol Endocrinol as a reference. First, this paper only refers to spermatogenesis, second, Persengiev et al. drew this conclusion after analysis of TBP protein expression. I think that this comment should be removed from the result part and possibly discussed properly in the discussion if really relevant.

Line 280: the TATA box and DPE are precisely located in relation to the transcription start site. If the TsTbp1 DPE is positioned at the right distance, the two not so well conserved TATA-like boxes at -97 and -67 are not. So, it is already clear that they are not likely to be functional as the 5’ RACE showed only 1 band. I would rephrase this paragraph. The EMSA in Figure 4 are actually a confirmation of these observations.

Line 295: the authors cloned the cDNAs encoding TsTAF6 and TsTAF9: the sequences are in supplementary figures but it would be very informative to show the alignment of these proteins with human TAF6 and TAF9 to highlight the conservation. As the antibodies used were raised against the human proteins, the specificity should be demonstrated: there is no direct evidence that the bands detected on Figure 5A are indeed the right ones. Similarly, the immunofluorescence experiments are not convincing because the specificity of the antibodies used is not established. The quality of the pictures is not very good (why the scale is different between the lower and upper panels?) and the localization close to the nuclear envelop is not an indication of active transcription as it is where heterochromatin is localized. What is the localization of TsTBP1? The observation that TsTAF6 and TsTAF9 do not always overlap is puzzling as there are multiple studies showing that TAF6 and TAF9 form a heterodimer that is incorporated in the core-TFIID complex in the cytoplasm during TFIID assembly, prior to the nuclear import of the complex (Wright PNAS (2006), Bieniossek Nature (2013)), so it is not clear whether free TAF6 or TAF9 really exist.

Line 320: in this paragraph, the authors present EMSA experiments and conclude that TsTAF6 and TsTAF9 are interacting with the DPE of the TsTbp1 promoter. These conclusions do not take into account the fact that TAF6/TAF9 is a building block within the TFIID complex, so a direct interaction is not demonstrated here. There are some studies where the TAF6 and TAF9 DPE binding domains have been mapped by EMSA using proteins produced in bacteria (Shao Mol Cell Biol 2005): are these domains conserved in the TsTAF6 and TsTAF9 counterparts? To demonstrate a direct interaction, the authors should use recombinant proteins.

Line 358: As the sequences of TsTAF6 and TsTAF9 not identical to the human proteins, what is the structure predicted for the TsTAF6/TsTAF9 heterodimer by Alphafold, without using the hTFIID structure as a model?

Line 374: In the different published human TFIID structures, the structured domains are located far away from the DNA, as mentioned by the authors, so it is not clear how TAF6/TAF9 contact DNA in the structure. The authors showed that the TsTAF6/TsFA9 dimer display a positive patch. Is this positive patch also present in the hTAF6/hTAF9 dimer?

Line 380: I do not understand the molecular dynamics simulation (it is not my field, maybe it could be explained for a more general audience) and it is not clear to me how long the DPE probe is: is it 6 bp as indicated in figure 8A? How these data fit with the published structure showing that the DNA is quite rigid and not making contact with the structured domains? Shao et al. actually reported that the DPE interacting domain are not within the HFDs of TAF6 and TAF9.

Minor comments:

The interest of the study of cestodes in the context of the transcription initiation is not explained in the introduction.

In the material and methods, the references of the various antibodies are missing, including the one used against the human TBP used in figure 5A that is not mentioned.

The authors should mention the WormBase ParaSite database, as it contains already the information about TsTAF6 and TsTBP1 (protein sequences, genomic organization).

Line 34: NF1, YY1 and AP1 are not general transcription factors

Line 56: the sentence is incomplete

Line 60: this sentence needs to be rephrased

Line 63: more recent genome-wide analyses report less than 10-20% of TATA-box containing promoters in human: Gershenzon and Ioshikhes Bioinformatics (2005), Cooper et al. Genome Research (2006)

Line 66: this sentence needs to be rephrased

Line 69: this sentence needs to be rephrased

Line 72: I would recommend to tune down (if not remove) the reference to the TFIID enzymatic activities as it is disputed (Timmers, BBA – gene regulatory mechanisms (2021). Moreover, references are missing in this part of the paragraph.

Line 76: this sentence is not clear to me

Line 77: “s” is missing in organism-s

Line 224: add “genomic” for clarity

Line 225: TBP, not TBP1

Line 247: Human TBP not human Tbp1

Lines 291 and 292: TsTbp1 should be in italics

Line 296: I would recommend a transition sentence to explain why they identified tsTAF6 and TsTAF9.

Line 313: anti-human TBP

Line 347: not all TFIID structure has been resolved, so I would recommend to remove “full”

Figure 1: I would recommend to label the DPE and TATA sites rather than TAF6/9 and TBP

Figure 3B: it would be easier to write the restriction enzymes names

Figure 3D: the authors performed Student test with n=3. As there is not enough data to test the normal distribution, a non-parametric test should be used (however, with n=3, a Mann&Whitney test is always non significant…)

Figure 4D: Is it an anti TBP or an anti TsTBP1 antibody used in this supershift assay?

Figure 6: why is there no competition with 25x TsTbp1-DPE cold probe (even a slight increase in the signal intensity?)

6. PLOS authors have the option to publish the peer review history of their article (what does this mean?). If published, this will include your full peer review and any attached files.

Reviewer #1: No

Reviewer #2: No

Reviewer #3: No

---

## [Author Response · Author response to Decision Letter 0]

16 May 2024

Review Comments to the Author

We thank editor and the reviewers for your valuable time and comments for our article. These led to substantially improving the current version.

RESPONSE TO REVIEWER 1

Reviewer #1:

In this manuscript, the authors examined the promoter of the TBP gene (Tstbp1) in Taenia solium. By using 5'-RACE with total RNA from T. solium cysticerci, they identified the transcription start site (TSS) (Fig. 3A). Examination of the promoter sequence suggested the presence of a DPE motif at the unconventional +27 position relative to the A+1 (Fig. 1). The Tstbp1 promoter appears to lack a TATA box at the normal position (roughly -30 relative to the TSS), but there are TATA-like sequences at positions -97 and -69 relative to the TSS. Analysis of these TATA-like sequences by gel shift experiments with T. solium nuclear extracts did not reveal any DNA-binding activity (Fig. 4).

Because of the existence of a potential DPE in the Tstbp1 promoter, the authors investigated TAF6 and TAF9 in greater detail, as it had been previously found that TAF6 and TAF9 can crosslink and potentially bind to the DPE. Western blot analysis of T. solium nuclear extracts with anti-TsTBP1, anti-human TAF6, and anti-human TAF9 revealed bands that appear to correspond to the cognate proteins. Immunofluorescence of cysticerci with the anti-human TAF6 and TAF9 antibodies yielded images that may correspond to the cognate T. solium TAF6 and TAF9. Gel shift analyses with T. solium nuclear extracts and wild-type or mutant putative DPE sequences yielded bands that gave an apparent supershift upon addition of anti-human TAF6 as well as anti-human TAF9 (Fig. 6). Molecular modeling of TsTAF6 and TsTAF9 was also performed (Figs. 7 and 8), and suggested that TsTAF6-TsTAF9 dimers might interact with the putative DPE.

The authors' efforts to analyze the core promoter of the TBP gene in a challenging and important organism are appreciated. There are, however, too many unknowns to be able to recommend this manuscript for publication in PLoS ONE. Some of the issues are as follows.

1. First, it is not known that the putative DPE-like motif in the promoter is a functionally active DPE. Some sort of transcription assay with wild-type versus mutant DPE versions of the promoter should be performed to assess whether or not the DPE-like motif is important for core promoter activity. In addition, it is questionable whether the sequence is a genuine functional DPE, as it is located at the +27 position rather than the standard +28 position. Most importantly, it must be determined whether or not the putative DPE is a functionally active DPE.

Normally the position of DPE element in promoters from different species starts at +28. However, the analysis of Tstbp1 core promoter showed a better similarity score to the RGWYV (consensus DPE) sequence to the motif +27TGTCG+31 (score of 4) instead of +28GTCGA+32 (score of 2). For that reason, we proposed the first motif as a putative DPE in Tstbp1 core promoter. As DPE has never been described in Cestodes, exist a possibility that the position of some core promoter elements is not as rigorous compared to other organisms. 

We used the following algorithm to determine similarity score:

# Define the degenerate sequence and the sequences to compare

degenerate = ['R', 'G', 'W', 'Y', 'V']

sequence_1 = ['T', 'G', 'T', 'C', 'G']

sequence_2 = ['G', 'T', 'C', 'G', 'A']

# Map degenerate nucleotides to their possible values

degenerate_to_nucleotides = {

 'R': ['A', 'G'],

 'G': ['G'],

 'W': ['A', 'T'],

 'Y': ['C', 'T'],

 'V': ['A', 'C', 'G']

}

# Function to calculate the similarity score

def calculate_score(degenerate, sequence, mapping):

 score = 0

 for deg, nucleotide in zip(degenerate, sequence):

 if nucleotide in mapping[deg]:

 score += 1

 return score

# Calculate scores for both sequences

score_1 = calculate_score(degenerate, sequence_1, degenerate_to_nucleotides)

score_2 = calculate_score(degenerate, sequence_2, degenerate_to_nucleotides)

score_1, score_2

Result

(4, 2)

Additionally, we tested the luciferase activity of constructs containing the core promoter of Tstbp1, as well as constructs where the DPE and Inr were mutated. A 30% decrease in luciferase activity was observed when the DPE was removed, and an 80% decrease when both, Inr and DPE, were removed (activity similar to the empty vector). 

It suggests the functionality of the proposed DPE, as well as of the Inr. This new result has been added as the Figure 4 in the manuscript. Additionally, we added the methodology in the material and methods section for Plasmid constructions and luciferease reporter assay (Lines 178 – 190) As well we added the analysis in the result and discussion section (Lines 331 – 345).

2. The gel shift analysis in Fig. 6 does not provide any evidence whether TAF6 and/or TAF9 interact directly with the DPE sequence. The shifted bands seen with the nuclear extract could be due to the binding of non-TAF6/9 subunits of a multiprotein complex (such as TFIID) that also contains TAF6 and TAF9. In that case, the antibodies against TAF6 and TAF9 would cause a supershift. It might be possible to clarify this point by carrying out the gel shift analyses with purified TsTAF6 and TsTAF9 proteins.

We believe that the gel shift assay in Fig. 6 shows evidence of the formation of a complex including TsTAF6, TsTAF9 and the DPE-probe, due to the presence of a super shifted band when anti-TAF6 or anti-TAF9 was added to the reaction. Certainly, the bands observed in Fig. 6 can be a multiprotein complex (TFIID) that includes TsTAF6 and TsTAF9, that are located in some position that allows the antibodies to recognize them. We agree that gel shift assays using recombinant purified TsTAF6 and TsTAF9 proteins would be direct evidence. However, we have been facing problems to produce these recombinant proteins in the laboratory, for that reason, it was not possible to include it in this manuscript. 

Therefore, we performed molecular modelling, followed by molecular dynamic simulations in order to evaluate the possibility of: 1) the formation of a TsTAF6-TsTAF9 complex (as observed in other organisms) and 2) the formation of a ternary complex TsTAF6-TsTAF9-DPE-probe. Our results suggesting the possibility of a contact of both TAFs (TsTAF6-TsTAF9) to DNA (Tstbp1-DPE-probe).

Reviewer #2: 

The work of Rodriguez-Lima et al. aims to expand our understanding of the transcription machinery in cestodes. TFIID is a general transcription factor found across species that consists of the TATA binding protein (TBP) and a set of highly conserved TBP associated factors (TAFs). The Landa lab previously cloned the cDNA for the Taenia solium TATA binding protein (TsTBP1). This manuscript builds on that work by characterizing the TsTBP1 gene structure, regulatory elements and expression levels using standard techniques.

The author's more significant contribution is the identification and characterization of two addition TFIID subunit in Taenia solium, TAF6 and TAF9. Antibodies against human TAF6 and TAF9 proteins were used in westerns blots to identify proteins of the predicted molecular weight based on the cDNA sequence. The antibodies also were used in confocal microscopy to localize the proteins to the nucleus. What is the evidence that the antibodies against human TAF6 and TAF9 antibodies will recognize the Taenia solium proteins in western blots? A figure that illustrates the degree amino acid sequence homology between the human and Taenia solium protein would be helpful. The authors might consider expressing TsTAF6 and TsTAF9 and verify the cross-reactivity of the antibodies being used for these studies.

A supplementary figure 3 was added to the manuscript. This figure shows alignment comparing primary sequences from Taenia solium (Ts) and Homo sapiens (Hs) of TAF6 and TAF9. The figure also shows the region used to produce the antibodies according to manufacturer, immunological analysis shows a conservation of antigenic epitopes presented in some regions in both species. Therefore, we used these antibodies for our experiments. See lines 379 -382 in main text.

TAF6 and TAF9 have been shown to bind to the downstream promoter element (DPE), which was first discovered downstream of transcription start sites in drosophila. The authors set out to investigate if this also is the case in Taenia solium using EMSA, homologous competition and human TAF6 and TAF9 antibodies. As shown in Figure 6, mobility shifted complexes are detected and decrease in intensity with the addition of unlabeled competitor probe. Once again their is concern about the specificity of antibodies being used for the supershift experiments. The appearance of the supershifted complex due to the binding of the TAF6 or TAF9 antibody is not accompanied by any detectable decrease in the TAF6- or TAF9-DNA complexes.

As previously mentioned, the antibodies are recognizing the transcription factors in both species. On the other hand, we repeat the EMSA experiment and replace the figure where we demonstrated the supershift bands and the decrease on intensity of shifted bands when antibody was added. Moreover, we reviewed this experiment, and detected an error in the amount of nuclear extract added in previous EMSA (see the new figure 7 and modification of text in lines 404 - 406).

The last sentence in the abstract states "Novel and interesting features of the TsTAF6-TsTAF9 complex for interaction with DPE on T. solium promoters are discussed." The modeling of TAF6-TAF9 dimer binding to DNA is a powerful approach and great starting point. However, it was not clear what new information was uncovered that increased our understanding about the interaction of TAF6 and TAF9 with promoter DNA.

Modeling of the TsTAF6-TsTAF9 complex reveals the presence of prominent positively charged regions on the surface of the HFD pair and HEAT repeat domains. This suggesting a DNA recognition function, as occurs in the nucleosomal histone octamer. Importantly, the surfaces of the A and B lobes in Homo sapiens TFIID lack significant positive patches. The novel feature is described in Results and Discussion section (lines 453 – 459), but we expanded the description to be clearer in conclusions section, lines 514 – 517.

Reviewer #3: PONE-D-23-43961

In this manuscript, Rodriguez-Lima et al. reports the characterization of the Taenia solium cestode TBP gene (TsTbp1) and of its promoter which contains 2 TATA-like elements, an Inr and one DPE element. They showed that the TATA-like elements are not bound by TsTBP1. They cloned the genes encoding TsTAF6 and TsTAF9 and reported an interaction between these proteins and TsTbpP1 DPE element by EMSA. They build a model of the interaction between TsTAF6, TsTAF9 and DPE DNA.

I have some concerns that the data are interesting but the potential novelty is not really highlighted. There are already lots of data published on Pol II transcription initiation and the cloning and characterization of the TAFs in yeast, insects or mammals were carried out several decades ago. The authors should explain more what their new cestode model brings to the Pol II transcription initiation field.

We added more information on the importance of studying promoters on model interaction in the introduction section (see lines 88 – 97). Moreover, in the conclusions section we highlight the novelty of the work (Lines 514 – 517).

Figure 2: The authors compare the genomic structure of the Tbp gene in different species. I do not understand the interest of this figure and I am not convinced by their conclusion that “the intron position and gene structure in TsTbp1 are similar to those observed in human Tbp1, highlighting the importance of this transcription factor across different organisms”: the human Tbp gene has 7 exons, TsTbpl1 only 5, the size of the exons is not similar, the position of the introns is not similar… To me, the real conclusion of this figure is that there is a huge discrepancy in the organization of these orthologs…

Thank you for your comment. We agree with it. We have modified the text, please see results section, lines 298 - 301 on the manuscript.

Figure 3B: It is not clear why the intensity of the upper band is weaker than the lower band. Is the BglII site located close to the extremity of the region recognized by the probe? It would be informative to have a map of the locus with the restriction sites (at least for BglII). The conclusion of the authors is that TsTbp1 is present in a single copy. This need to be clarified: does this mean that there is no Tbp paralog? As Trf2/Tbpl1 is present in metazoans, including C elegans, if true, this would be an interesting observation. The hybridization conditions are not described in the material and methods and it is not possible to know whether they are stringent or not.

You right about it, the Bgl II site is localized end of final exon position 1097 of gene with respect to TSS of TsTBP1. Because the region that hybridize is shorth, it may be the reason why the band is weak. On the other hand, we found one paralog gene to TsTBP (Trf2, TsM_000321300) in the genome project T. solium Data bank. We obtained a specific restriction pattern for Tstbp1 gene. In this paralog, there are no sites for Bgl II. Moreover, there are EcoRI and Hind III sites outside of gene which will give sizes of approximately 4 and 5 kb. However, no bands were observed in the Southern blot. But as we used a random primer labeled kit and hybridization condition highly stringent, none unspecific bands were observed. Therefore, we only identified of Tstbp1 gene (please, see Southern blot section in methods section). The hybridization conditions were added to the manuscript in material and methods section, lines 157 -162.

In Figure 3D, the authors used SOD6 as a normalizing gene for the RT-qPCR performed on cysticerci and adults. This is not a classical housekeeping gene used (at least in other models…): what is the evidence that it is a good marker? Are the data comparable when using Gapdh or 18S RNA?

We have never used 18S RNA. We previously tested the amplification of several genes that encode proteins in order to find a good control to housekeeping gene for our expression studies in cysticerci and adults (Data not published). We have found that Cu/Zn-SOD is a good one, and we have already used it in 2 publications (Jiménez et al., 2015 and Miranda et al., 2024). Regarding the use of GAPDH, a recent pre-print publication shows that its expression is variable in different stages of this parasite (Gutierrez-Loli et al (2022, doi: https://doi.org/10.1101/2022.03.22.485324), that was the reason for not using GAPDH gene.

Line 267: from the RT-qPCR experiment, the authors conclude that TsTBP1 might be involved in growth and egg production in the adult stage, citing Persengiev (1996) Mol Endocrinol as a reference. First, this paper only refers to spermatogenesis, second, Persengiev et al. drew this conclusion after analysis of TBP protein expression. I think that this comment should be removed from the result part and possibly discussed properly in the discussion if really relevant.

You right, we have modified the phrase (See line 320 – 321). 

Line 280: the TATA box and DPE are precisely located in relation to the transcription start site. If the TsTbp1 DPE is positioned at the right distance, the two not so well conserved TATA-like boxes at -97 and -67 are not. So, it is already clear that they are not likely to be functional as the 5’ RACE showed only 1 band. I would rephrase this paragraph. The EMSA in Figure 4 are actually a confirmation of these observations.

We fixed the paragraph, see lines 354 – 360.

Line 295: the authors cloned the cDNAs encoding TsTAF6 and TsTAF9: the sequences are in supplementary figures but it would be very informative to show the alignment of these proteins with human TAF6 and TAF9 to highlight the conservation. As the antibodies used were raised against the human proteins, the specificity should be demonstrated: there is no direct evidence that the bands detected on Figure 5A are indeed the right ones. Similarly, the immunofluorescence experiments are not convincing because

---

## [Decision Letter · Decision Letter 1]

21 Jun 2024

Taenia solium TAF6 and TAF9 bind to a downstream promoter element present in the Tstbp1 gene core promoter

PONE-D-23-43961R1

Dear Dr. Landa,

We’re pleased to inform you that your manuscript has been judged scientifically suitable for publication and will be formally accepted for publication once it meets all outstanding technical requirements.

Please note the minor comments of reviewers #1 and #3 and consider addressing them.

Kind regards,

Tamar Juven-Gershon, Ph.D.

Academic Editor

PLOS ONE

Additional Editor Comments (optional):

Reviewers' comments:

Reviewer's Responses to Questions

**Comments to the Author**

1. If the authors have adequately addressed your comments raised in a previous round of review and you feel that this manuscript is now acceptable for publication, you may indicate that here to bypass the “Comments to the Author” section, enter your conflict of interest statement in the “Confidential to Editor” section, and submit your "Accept" recommendation.

Reviewer #1: All comments have been addressed

Reviewer #2: All comments have been addressed

Reviewer #3: All comments have been addressed

2. Is the manuscript technically sound, and do the data support the conclusions?

Reviewer #1: Yes

Reviewer #2: Yes

Reviewer #3: Yes

3. Has the statistical analysis been performed appropriately and rigorously? 

Reviewer #1: Yes

Reviewer #2: Yes

Reviewer #3: Yes

4. Have the authors made all data underlying the findings in their manuscript fully available?

Reviewer #1: Yes

Reviewer #2: Yes

Reviewer #3: (No Response)

5. Is the manuscript presented in an intelligible fashion and written in standard English?

Reviewer #1: Yes

Reviewer #2: Yes

Reviewer #3: Yes

6. Review Comments to the Author

Reviewer #1: The authors have suitably addressed my comments. The authors might note that the DPE (extended version is called the DPR) is sometimes found at the +27 position (rather than the canonical +28 position) in humans [Vo ngoc et al, Genes Dev. 37: 377-382 (2023)]. The same might be true for Taenia.

Reviewer #2: (No Response)

Reviewer #3: I have just two comments:

Figure 2 and lines 298-301 : the text has been modified but I am not sure it is very clear. There is no conservation of the genomic organization of the gene (lines 298-299): I do not understand what does it mean that “structural organization is not relevant for this TF”, this should be rephrased/clarified.

Line 351: indicate the post hoc test used after the ANOVA in Figure 4

7. PLOS authors have the option to publish the peer review history of their article (what does this mean?). If published, this will include your full peer review and any attached files.

Reviewer #1: No

Reviewer #2: No

Reviewer #3: No

---

## [Editor Report · Acceptance letter]

20 Aug 2024

PONE-D-23-43961R1 

PLOS ONE

Dear Dr. Landa, 

I'm pleased to inform you that your manuscript has been deemed suitable for publication in PLOS ONE. Congratulations! Your manuscript is now being handed over to our production team.

Kind regards, 

on behalf of

Prof. Tamar Juven-Gershon 

Academic Editor

PLOS ONE